# Efficient Policy Evaluation Across Multiple Different Experimental Datasets

**Yonghan Jung***
Purdue University
jung222@purdue.edu

**Alexis Bellot***[†]
Independent Researcher
abellot@gmail.com

## Abstract

Artificial intelligence systems are trained combining various observational and experimental datasets from different source sites, and are increasingly used to reason about the effectiveness of candidate policies. One common assumption in this context is that the data in source and target sites (where the candidate policy is due to be deployed) come from the same distribution. This assumption is often violated in practice, causing challenges for generalization, transportability, or external validity. Despite recent advances for determining the identifiability of the effectiveness of policies in a target domain, there are still challenges for the accurate estimation of effects from finite samples. In this paper, we develop novel graphical criteria and estimators for evaluating the effectiveness of policies (*e.g.*, conditional, stochastic) by combining data from multiple experimental studies. Asymptotic error analysis of our estimators provides fast convergence guarantee. We empirically verified the robustness of estimators through simulations.

## 1 Introduction

In the empirical sciences, conclusions on the effect of actions or policies is often supported by evidence drawn from prior observations and experiments. The conditions under which such inferences can be formally justified can be traced back (in part) to Campbell, Stanley and Cook [10, 11, 14]. They argued for a basic dichotomy in the kinds of questions that scientists seek to answer from experimental data. On the one hand asking whether "*in fact, the experimental stimulus made some significant difference in this specific instance?*", and on the other hand asking "*to what populations, settings, and treatments can this effect be generalized?*" [10, p. 297]. These inferences have since been labelled as *internal validity* and *external validity*, respectively.

External validity is concerned with the extent to which findings from one population can be "re-processed", or "re-calibrated" so as to circumvent population differences and produce valid generalizations in a target population where experiments cannot be performed (e.g., outside the laboratory, different domains, etc.). The validity of these inferences will necessarily be contingent on a careful analysis to ascertain the commonalities and differences between domains as, for example, if the target domain is completely arbitrary generalization is impossible. In the causal transportability literature, the basis for generalization (also called transportability) is justified by the stability and invariance of the causal mechanisms shared across populations and domains [20, 32]. Several graphical characterizations exist to delineate the conditions under which transportability is possible, with recent algorithms proposing solutions for general instances of the external validity task combining observational and experimental distributions under partial observability [36, 2, 3, 16, 30].

---

*Equal contribution. Correspondence to Yonghan Jung: jung222@purdue.edu.

[†]Now at Google DeepMind. Work done partly while affiliated with Columbia University and partly in an independent capacity.

38th Conference on Neural Information Processing Systems (NeurIPS 2024).

These algorithmic solutions express a target policy effect in terms of the observational and experimental source distributions. Still, then one needs to go further and estimate the resulting expression from finite samples. In practice, with a finite number of samples and potentially high-dimensional covariates, estimating causal expressions is quite challenging. Effective estimators have been developed for specific settings, starting with doubly-robust estimators for functionals given by the backdoor criterion [13, 37, 9, 45], and recently extended to cover general identification scenarios with observational and experimental samples [25, 26, 8]. These techniques also find parallels across other related disciplines, such as reinforcement learning where re-weighting [42, 31], outcome modelling [6], and doubly-robust estimation [18], are common for evaluating the effect of policies to overcome shifts in the behaviour policy. Recently, [44] and [22] have considered policy evaluation under covariate shift and selection bias, a special case of the external validity problem with a given graph. Despite their generality, existing estimators still only cover a limited portion of realistic scientific inferences. In particular, existing methods are not applicable in settings where datasets are collected in different domains.

We consider the generalization of causal claims from observational and experimental data through the task of *policy evaluation*. The target for inference is $\mathbb{E}_{P_\pi^0}[Y]$ where $P_\pi^0$ symbolizes the distribution of data in a target domain (indexed as 0) in which a hypothetical policy of interest $\pi$ (also known as dynamic treatment regimes [33] or soft interventions [16]) has been implemented. The question then becomes how to identify and estimate $\mathbb{E}_{P_\pi^0}[Y]$, given finite samples from multiple observational and experimental data (e.g., $P_{\pi_i}^i$, a source domain indexed by $i$ in which experimental policy is $\pi_i$ that may differ with $\pi_0$) collected under different settings and structural assumptions, encoded in causal diagrams. We aim to bridge the gap between identification and estimation to solve general instances of external validity. Our contributions are twofold:

1. **Sec. 3:** We develop nonparametric identification criteria (Thm. 1) to determine whether the effect of a policy may be expressed through an adjustment formula from two separate distributions induced by policy interventions, collected from different populations. Based on this formulation, we develop a multiply robust estimator (Thm. 3) that enjoys multiply robustness against model misspecification and bias.

2. **Sec. 4:** We generalize these identification criteria (Thm. 4) and propose a general multiply-robust estimator (Thm. 6) applicable for the evaluation of policies from multiple source datasets.

## 1.1 Preliminaries

We use bold letters ($\mathbf{X}$) to denote a random vector and $X$ a random value. Each random vector is represented with a capital letter ($\mathbf{X}$) and its realized value with a small letter ($\mathbf{x}$). Given a set $\mathbf{X} = \{X_1, \cdots, X_n\}$, we denote $\mathbf{X}^{(i)} := \{X_1, \cdots, X_i\}$. For a discrete vector $\mathbf{X}$, we use $\mathbb{1}_{\mathbf{x}}(\mathbf{X})$ to represent the indicator function such that $\mathbb{1}_{\mathbf{x}}(\mathbf{X}) = 1$ if $\mathbf{X} = \mathbf{x}$; $\mathbb{1}_{\mathbf{x}}(\mathbf{X}) = 0$ otherwise. For comprehensibility, we use $P(\mathbf{v})$ to denote a probability at $\mathbf{V}$ at $\mathbf{v}$ for discrete/continuous random variables $\mathbf{V}$. In similar, we use $\sum_{\mathbf{z}}$ for $\mathbf{Z} \subseteq \mathbf{V}$ for the summation/integration over a mixture of discrete/continuous random variables $\mathbf{Z}$ For example, we write the back-door adjustment as $\sum_{\mathbf{z}} \mathbb{E}_P[Y \mid x, \mathbf{z}] P(\mathbf{z})$ even when $\mathbf{Z}$ is a mixture of discrete/continuous variables. We use $\mathbb{E}_P[f(\mathbf{V})] := \sum_{\mathbf{v}} f(\mathbf{v}) P(\mathbf{v})$ for a function $f$. For a sample set $\mathcal{D} := \{\mathbf{V}_{(i)} : i = 1, \cdots, n\}$ where $\mathbf{V}_{(i)}$ denotes the $i$th samples, we use $\mathbb{E}_{\mathcal{D}}[f(\mathbf{V})] := (1/n) \sum_{i=1}^n f(\mathbf{V}_{(i)})$. We use $\|f\|_P := \sqrt{\mathbb{E}_P[\{f(\mathbf{V})\}^2]}$. If a function $\hat{f}$ is a consistent estimator of $f$ having a rate $r_n$, we use $\hat{f} - f = o_P(r_n)$. We say $\hat{f}$ is $L_2$-consistent if $\|\hat{f} - f\|_P = o_P(1)$. We use $\hat{f} - f = O_P(1)$ if $\hat{f} - f$ is bounded in probability, and $\hat{f} - f = O_P(r_n)$ when $\hat{f} - f$ is bounded in probability at rate $r_n$.

We use Structural Causal Models (SCMs) as our framework [35]. An SCM $\mathcal{M}$ is a quadruple $\mathcal{M} = \langle \mathbf{U}, \mathbf{V}, P(\mathbf{U}), \mathcal{F} \rangle$. $\mathbf{U}$ is a set of latent variables following a joint distribution $P(\mathbf{U})$. $\mathbf{V}$ is a set of observable variables whose values are determined by functions $\mathcal{F} = \{f_{V_i} : V_i \in \mathbf{V}\}$ such that $V_i \leftarrow f_{V_i}(\mathbf{pa}_{V_i}, \mathbf{u}_{V_i})$ where $\mathbf{PA}_i \subseteq V$ and $\mathbf{U}_{V_i} \subseteq \mathbf{U}$. Each SCM $\mathcal{M}$ induces a distribution $P(\mathbf{V})$ and a causal graph $\mathcal{G}$ in which directed edges from every variable in $\mathbf{PA}_i$ to $V_i$ exist. Dashed-bidirected arrows encode correlated latent variables.

## 2 Policy Evaluation Integrating Multiple Experimental Datasets

We investigate the sequential decision-making setting concerning a set of actions $\mathbf{X}$, a series of dynamic covariates $\mathbf{Z}$, a series of static covariates $\mathbf{C}$ and an outcome variable of interest $Y$ in an SCM $\mathcal{M}$. A policy vector $\boldsymbol{\pi} := \{\pi^i\}$ over actions $\mathbf{X} = \{X_1, \cdots, X_m\}$ is an ordered set of decision rules for each $X_i \in \mathbf{X}$. Actions are selected according to a topological ordering $X_1 \prec \cdots \prec X_K$ over time. Each action $X_i$ is potentially associated with a set of prior static and dynamic covariates, for example, the decision rule for $X_k$ could be defined as $x_k \sim \pi(\cdot \mid \mathbf{z}^{(k)}, \mathbf{x}^{(k-1)}, \mathbf{c}^{(k)})$. Every $\pi(X_k \mid \mathbf{Z}^{(k)}, \mathbf{X}^{(k-1)}, \mathbf{C}^{(k)})$ is a probability distribution mapping from domains of the set of inputs $\{\mathbf{Z}^{(k)}, \mathbf{X}^{(k-1)}, \mathbf{C}^{(k)}\}$ to the domain of actions $X_k$. The implementation of a policy $\pi$ in $\mathcal{M}$ induces an intervened model $\mathcal{M}_\pi$, that sets values of every $X \in \mathbf{X}$ to be decided by the policy $\pi$, replacing the functions $\{f_X, X \in \boldsymbol{X}\}$ that would normally set its value. We denote a distribution induced by $\mathcal{M}_\pi$ as $P_\pi$. Now, we fix the notion of the *policy evaluation* as follows:

**Definition 1** (**Policy evaluation** [41]). *The policy evaluation is to predict the effectiveness of a policy vector $\pi$ on an outcome $Y$ in an target SCM $\mathcal{M}^0$; i.e., $\psi_0 := \mathbb{E}_{P_\pi^0}[Y]$.*

Difficulties in estimating $\mathbb{E}_{P_\pi^0}[Y]$ comes from that the distribution or samples from $P_\pi^0$ are generally not available. These discrepancies can be formalized under the rubric of SCMs as follows. In the most general setting, an investigator might leverage multiple source domains $\{\mathcal{M}^1, \mathcal{M}^2, \ldots, \mathcal{M}^K\}$ over $\mathbf{V}$ that entail distributions $\mathbb{P} : \{P^1, P^2, \ldots, P^K\}$. Data or samples from these distributions may be available under different behaviour policies, *e.g.*, $\pi_1, \pi_2, \ldots, \pi_K$, depending on the study or data collection protocol implemented in each domain (that might include an observational regime, *i.e.* no policy implemented). To ground the policy evaluation problem, we define graphical tools to capture commonalities and discrepancies across domains.

**Definition 2** (**Domain discrepancy** [29]). *For every pair of SCMs $\mathcal{M}^i, \mathcal{M}^j$ ($i, j \in \{0, 1, 2, \ldots, K\}$) defined over $\mathbf{V}$, the domain discrepancy set $\Delta_{ij} \subseteq \mathbf{V}$ is defined such that for every $V \in \Delta_{ij}$ there might exist a discrepancy between $f_V^{M^i} \neq f_V^{M^j}$, or $P^{M^i}(\boldsymbol{u}_V) \neq P^{M^j}(\boldsymbol{u}_V)$.*

**Definition 3** (**Selection diagram** [29]). *The selection diagram $\mathcal{G}^\Delta = \{\mathcal{G}^j\}_{j \in \{0,1,2,\ldots,T\}} \cup \{\mathcal{G}^{\Delta_{0j}}\}_{j \in \{1,2,\ldots,T\}}$ is a graph constructed from $\mathcal{G}^i$ ($i \in \{0, 1, 2, \ldots, T\}$) by adding the selection node $S_{ij}$ to the vertex set, and adding the edge $S_{ij} \to V$ for every $V \in \Delta_{ij}$.*

$\Delta_{i,j}$ locates the mechanisms where structural discrepancies between two domains are suspected to take place. $V \notin \Delta_{i,j}$ represents the assumption that the mechanisms for $V$ are invariant across the two domains. The induced selection diagram is a parsimonious representation of these constraints. The following example illustrates these notions.

**Example 1** (**External validity under covariate shift**). A common instance of the external validity problem in the literature considers the evaluation the effect of a policy $\pi : \Omega_C \times \Omega_X \to [0, 1]$ for assigning a treatment $X \in \{0, 1\}$, subject to shift in the distribution of covariates $C$. For this example, let source and target domains $\mathbb{M} : \{\mathcal{M}^1, \mathcal{M}^0\}$ over $\boldsymbol{V} = \{C, X, Y\}, \boldsymbol{U} = \{U_{XY}, U_C\}$ be defined as follows,

$$
\mathcal{M}^1 : \begin{cases} \mathcal{F} & = \begin{cases} C \leftarrow f_C(U_C) \\ X \leftarrow f_X(C, U_{XY}) \\ Y \leftarrow f_Y(X, C, U_{XY}) \end{cases} \\ P(\boldsymbol{U}) & = P(U_{XY})P(U_C) \end{cases} \qquad \mathcal{M}^0 : \begin{cases} \mathcal{F}^0 & = \begin{cases} C \leftarrow f_C^0(U_C) \\ X \leftarrow f_X(C, U_{XY}) \\ Y \leftarrow f_Y(X, C, U_{XY}) \end{cases} \\ P^0(\boldsymbol{U}) & = P(U_{XY})P^0(U_C) \end{cases}
$$

Here, $C \in \Delta_{1,0}, \{Y, C\} \notin \Delta_{1,0}$ as only the mechanism for $C$ varies across domains. Consider the evaluation of $\pi : \pi(X = 1 \mid c) := 1/(1 + \exp\{-c\})$ given an experimental dataset in $\mathcal{M}^1$ in which $X$ has been randomized, *i.e.*, $X \sim \text{Bern}(0.5)$, and covariate data $P^0(C)$ available in $M^0$. Notice that we do not have access to the specification of the SCMs $\mathbb{M}$, but only the induced diagrams $\mathcal{G}^\Delta$, and a subset of entailed distributions $\mathbb{P} : \{P_{\text{rand}(X)}^1(X, Y, C), P^0(C)\}$. The policy effect is expressible as

$$
\mathbb{E}_{P_\pi^0}[Y] = \sum_{x,c,y} y P_{\text{rand}(x)}^1(y \mid c, x)\pi(x \mid c)P^0(c),
$$

and estimated given the policy $\pi$ and the combination of the available data from $P^1, P^0$. ∎

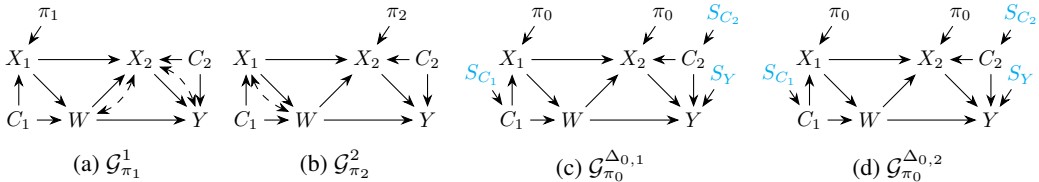

Figure 1: Graphs illustrating the inference of a two-stage treatment strategy $\pi_0 := \{\pi_0(x_1 \mid c_1), \pi_0(x_2 \mid x_1, c_2, w)\}$ given data from source domains $\mathcal{M}^1, \mathcal{M}^2$, described in Example 2.

## 3 Combining experiments from two domains

Example 1 illustrates two challenges in combining data from different domains to infer the effect of a new policy in a target domain. In a first instance highlighting the challenge of *identification*, that is inferring an expression in terms of $\mathbb{P}$ that identifies the policy effect, and in a second instance highlighting the challenge of *estimation*, that is providing efficient estimators from finite samples for the identified policy effect. The following example will serve to motivate this setting.

**Example 2** (**Two-stage treatment strategies**). A team of physicians is contemplating a treatment plan $\pi_0$ against heart disease $Y$ for their patients in $\mathcal{M}^0$. They consider administrating two drugs in sequence: a drug against hypertension $X_1$, followed by an anti-diabetic drug $X_2$ depending on the effect of $X_1$ on blood pressure $W$. To support their evaluation, two studies exist on these drugs, from domains $\mathcal{M}^1, \mathcal{M}^2$, that, however, have only analyzed their effect in isolation (on $X_1$ and $X_2$ separately) and under different treatment guidelines, $\pi_1, \pi_2$ respectively. The data collected refers to the variables $\mathbf{V} := (Y, \mathbf{C}_1, \mathbf{C}_2, X_1, X_2, W)$ in which $(\mathbf{C}_1, \mathbf{C}_2)$ are demographic variables. Formally, we assume physicians have access to $\mathbb{P} : \{P_{\pi_1}^1(\mathbf{V}), P_{\pi_2}^2(\mathbf{V}), P^0(\mathbf{C}_1, \mathbf{C}_2)\}$. The superscripts in $P^0, P^1, P^2$ are the index for the domain, and the subscripts $\pi_0, \pi_1, \pi_2$ denote the policies for assigning treatments. $\mathcal{G}^\Delta$ in Fig. 1 encodes the structural assumptions, which include discrepancies across domains and implemented policies in the available data. For example, the graph $\mathcal{G}_{\pi_1}^1$ specifies the known guideline $\pi_1$ used in $\mathcal{M}^1$, while no specific plan was followed for the assignment of $X_2$, that in practice depends on the patient's covariates $C_2$ as well as unobserved factors, e.g. mood, health awareness, etc. (summarized in the bi-directed arc). In addition, selection diagrams describe differences between domains. For example, the edge $\{S_{C_1} \to C_1\}$ in $\mathcal{G}_{\pi_0}^{\Delta_{0,1}}$ indicates a potential change in the distribution of covariates $C_1$ across domains $M^0, M^1$. The question then becomes how to estimate $\mathbb{E}_{P_{\pi_0}^0}[Y]$ given $(\mathcal{G}^\Delta, \mathbb{P})$. ∎

### 3.1 Identification

Example 2 illustrates the complexity of drawing inferences from multiple datasets collected under different settings. We extend this example to provide a general identification procedure for the effect of policies when two source datasets subject to different policies and/or discrepancies with the target domain are available. Let $\mathbf{V} := (Y, \mathbf{C}, X_1, \mathbf{W}, X_2, \mathbf{W}, Y, \mathbf{S})$ denote a set of disjoint variables, where $Y$ is an outcome variable, $\mathbf{C}, (\mathbf{C}, \mathbf{W})$ are covariates corresponding to two experiments, $(X_1, X_2)$ are treatment variables, and $\mathbf{S}$ denotes the selection nodes describing discrepancies across pairs of domains. Formally, the task signature is given as follows:

- **Input**: Samples from $\mathbb{P} = \{P_{\pi_1}^1(\mathbf{V}), P_{\pi_2}^2(\mathbf{V}), P^0(\mathbf{C}_1, \mathbf{C}_2)\}$; structural assumptions $\mathcal{G}^\Delta := \{\mathcal{G}_{\pi_0}^0, \mathcal{G}_{\pi_1}^1, \mathcal{G}_{\pi_2}^2, \mathcal{G}_{\pi_0}^{\Delta_{0,1}}, \mathcal{G}_{\pi_0}^{\Delta_{0,2}}\}$.

- **Query**: Estimate $\mathbb{E}_{P_{\pi_0}^0}[Y]$ where $P^0$ is distribution on the target domain and $\pi_0$ is a target policy assigning treatments with $\pi_0(X_1 \mid \mathbf{C}_1)$ and $\pi_0(X_2 \mid \mathbf{C}_2, W)$.

Given these inputs, a sufficient condition for identifying the query is given as follows:

**Definition 4** (**Adjustment criterion for combining two experiments**). *Given $\mathcal{G}^\Delta$, the adjustment criterion for combining two experimental datasets is defined by the following d-separation statements:*

1. ***Domain transfer for*** $Y$: $(Y \perp\!\!\!\perp \mathbf{S} \mid \mathbf{C}, X_1, X_2, W)$ *in* $\mathcal{G}_{\pi_0}^{\Delta_{0,2}}$; *i.e., the distribution over $Y$ is invariant between the source distribution from $\mathcal{M}^2$ and the target.*

2. **Domain transfer for** $W$: $(W \perp\!\!\!\perp \mathbf{S} \mid \mathbf{C}, X_1)$ in $\mathcal{G}_{\pi_0}^{\Delta_{0,1}}$; i.e., the distribution over $W$ is invariant between the source distribution from $\mathcal{M}^1$ and the target.

3. **Adjustment for** $Y$: $(Y \perp\!\!\!\perp \pi_i \mid C_1, C_2, X_1, X_2, W)$ in $\mathcal{G}_{\pi_i}$ for $i \in \{0, 2\}$; i.e., the distribution over $Y$ is invariant between regimes $\pi_0$ and $\pi_2$.

4. **Adjustment for** $W$: $(W \perp\!\!\!\perp \pi_i \mid C_1, C_2, X_1)$ in $\mathcal{G}_{\pi_i}$ for $i \in \{0, 1\}$; i.e., the distribution over $W$ is invariant between regimes $\pi_0$ and $\pi_1$.

The adjustment criterion could be shown to hold for Example 2. Specifically, domain transfer $Y$ could be shown by inspecting Fig.5e as the set $\mathbf{S} = \{S_{C_1}, S_{C_2}, S_W\}$ is d-separated from $Y$, conditional on $\{C_1, C_2, X_1, X_2, W\}$. Similarly, domain transfer for $W$ holds as $\mathbf{S} = \{S_{C_1}, S_{C_2}, S_Y\}$ is d-separated from $W$, given $C_1, C_2, X_1$ in Fig.5d. Similarly, one could verify the adjustment condition for $Y$ by inspecting Fig.5c and the adjustment condition for $W$ by inspecting Fig. 5a.

For this example, these conditions imply identifiability of the target query $\mathbb{E}_{P_{\pi_0}^0}[Y]$ given $(\mathcal{G}^\Delta, \mathbb{P})$.

**Theorem 1** (**Adjustment for combining two experiments**). *Under the adjustment criterion in Def. 4, the target query* $\psi_0 := \mathbb{E}_{P_{\pi_0}^0}[Y]$ *is identifiable from the samples from* $P_{\pi_1}^1(\mathbf{V})$, $P_{\pi_2}^2(\mathbf{V})$, $P^0(\mathbf{C}_1, \mathbf{C}_2)$. *Specifically, it's expressed as follows* [3]:

$$\mathbb{E}_{P_{\pi_0}^0}[Y] = \sum_{w, \mathbf{c}, \mathbf{x}} \mathbb{E}_{P_{\pi_2}^2}[Y \mid \mathbf{c}, w, \mathbf{x}] \pi_0(x_2 \mid \mathbf{c}, w) P_{\pi_1}^1(w \mid \mathbf{c}_1, x_1) \pi_0(x_1 \mid \mathbf{c}) P^0(\mathbf{c}), \tag{1}$$

*where* $\mathbf{X} := (X_1, X_2)$ *and* $\mathbf{C} := (\mathbf{C}_1, \mathbf{C}_2)$.

Effectively, despite the differences across domains encoded in Example 2, the effect of the new combination of anti-diabetic and anti-hypertensive drugs $\pi_0$, can be estimated using samples from experiments already conducted in $\mathcal{M}^1$, $\mathcal{M}^2$, and baseline characteristics of patients in $\mathcal{M}^0$.

## 3.2 Estimation

This section considers the estimation of the effect of policies, building on the identification criterion in Thm. 1. We first parameterize the identification estimand in Eq. (1) with two types of nuisance parameters $\boldsymbol{\mu}$ and $\boldsymbol{\omega}$. $\boldsymbol{\mu}$ is a collection of regression parameters, and $\boldsymbol{\omega}$ is a collection of the ratio of distributions.

The regression nuisance parameters are defined as follows: $\mu_0^2(\mathbf{C}, W, \mathbf{X}) := \mathbb{E}_{P_{\pi_2}^2}[Y \mid \mathbf{C}, W, \mathbf{X}]$ and $\breve{\mu}_0^2(\mathbf{C}, W, X_1) := \sum_{x_2} \mu_0^2(\mathbf{C}, W, X_1, x_2) \pi_0(x_2 \mid \mathbf{C}, W, X_1)$. Recursively, $\mu_0^1(\mathbf{C}, X_1) := \mathbb{E}_{P_{\pi_1}^1}[\breve{\mu}_0^2(\mathbf{C}, W, X_1) \mid \mathbf{C}, X_1]$ and $\breve{\mu}_0^1(\mathbf{C}) := \sum_{x_1} \mu_0^1(\mathbf{C}, x_1) \pi_0(x_1 \mid \mathbf{C})$. Eq. (1) can be parameterized as $\mathbb{E}_{P_{\pi_0}^0}[Y] = \mathbb{E}_{P^0}[\breve{\mu}_0^1(\mathbf{C})]$.

On the other hand, the ratio nuisance parameters $\omega_0^2, \omega_0^1$ are defined as functionals satisfying the following properties:

$$\mathbb{E}_{P_{\boldsymbol{\pi}}^0}[Y] = \mathbb{E}_{P_{\pi_2}^2}[\mu_0^2(\mathbf{C}, W, \mathbf{X}) \pi_0^2(\mathbf{C}, W, \mathbf{X})] = \mathbb{E}_{P_{\pi_1}^1}[\mu_0^1(\mathbf{C}, X_1) \pi_0^1(\mathbf{C}, X_1)]. \tag{2}$$

A closed form of the $\omega_0^i$ is provided in the later section at Eq. (14). By the definition of ratio nuisances, Eq. (1) can be parameterized as $\mathbb{E}_{P_{\pi_0}^0}[Y] = \mathbb{E}_{P_{\pi_2}^2}[\omega_0^2(\mathbf{C}, W, \mathbf{X}) Y]$. Equipped with these nuisances, we now present the DML-based estimator for the target query:

**Definition 5** (**DML for combining two experiments**). *Let* $\mathcal{D}^2 \sim P_{\pi_2}^2(\mathbf{V})$, $\mathcal{D}^1 \sim P_{\pi_1}^1(\mathbf{V})$ *and* $\mathcal{D}^0 \sim P^0(\mathbf{C})$. *Let* $L \geqslant 2$ *denote a fixed number.*

1. **Sample split:** *For* $\ell = 1, \cdots, L$, *randomly split* $\mathcal{D}^i$ *for* $i \in \{0, 1, 2\}$ *into L-fold. The* $\ell$*'th partition of the sample is denoted* $\mathcal{D}_\ell^i$. *The complement is* $\mathcal{D}_{-\ell}^i := \mathcal{D}^i \backslash \mathcal{D}_\ell^i$.

2. **Nuisance estimation:** *For each* $\ell = 1, \cdots, L$, *learn the estimator model* $\hat{\mu}_\ell^2$ *and* $\hat{\mu}_\ell^1$ *for* $\mu_0^2, \mu_0^1$ *using samples* $\mathcal{D}_{-\ell}^2, \mathcal{D}_{-\ell}^1$, *respectively. Also, learn the estimation model for* $\hat{\omega}_\ell^1, \hat{\omega}_\ell^2$ *for* $\omega_0^1, \omega_0^2$ *using samples* $\mathcal{D}_{-\ell}^i$ *for* $i = 0, 1, 2$, *respectively.*

---

[3]Thm. 1 remains valid for a mixture of discrete and continuous $W$, $\mathbf{C}$, and $\mathbf{X}$. For these cases, sums can be appropriately replaced by Lebesgue integrals. However, we continue to use summation notation in our explanation to keep the presentation of the identification result straightforward.

3. **Evaluation**: The DML estimator $\hat{\psi}$ for $\mathbb{E}_{P^0_{\pi_0}}[Y]$ is then given as

$$\hat{\psi} := \frac{1}{L} \sum_{\ell=1}^{L} \mathbb{E}_{\mathcal{D}^2_\ell}[\hat{\omega}^2_\ell \{Y - \hat{\mu}^2_\ell\}] + \mathbb{E}_{\mathcal{D}^1_\ell}[\hat{\omega}^1_\ell \{\check{\mu}^2_\ell - \hat{\mu}^1_\ell\}] + \mathbb{E}_{\mathcal{D}^0_\ell}[\check{\mu}^1_\ell]. \tag{3}$$

Estimating the ratio nuisance $\{\hat{\omega}^1, \hat{\omega}^2\}$ can be challenging due to the necessity of estimating density ratios like $\frac{P^1_{\pi_1}(\mathbf{C})}{P^2_{\pi_2}(\mathbf{C})}$ or $\frac{P^1_{\pi_1}(W|\mathbf{C}, X_1)}{P^2_{\pi_2}(W|\mathbf{C}, X_1)}$. We employ the classification-based method for estimating the density [17, Sec. 5.4]. To illustrate this method, consider estimating $\frac{P^1_{\pi_1}(\mathbf{C})}{P^2_{\pi_2}(\mathbf{C})}$. We assign $\lambda = 1$ if samples of $\mathbf{C}$ are from $P^1_{\pi_1}$ and $\lambda = 0$ if from $P^2_{\pi_2}$. Then, it's provable that $\frac{P^1_{\pi_1}(\mathbf{C})}{P^2_{\pi_2}(\mathbf{C})} = \frac{P(\lambda=1|\mathbf{C})}{P(\lambda=0|\mathbf{C})}$, which can be estimated using off-the-shelf probabilistic classification estimators.

The error of the DML estimator is presented below:

**Theorem 2** (**Learning Guarantees**). *Suppose* $\hat{\mu}^2_\ell, \hat{\mu}^1_\ell < \infty$ *and* $0 < \hat{\omega}^2_\ell, \hat{\omega}^1_\ell < \infty$. *Define* $\phi^2(\mathbf{V}; \mu^2, \pi^2) := \omega^2(\mathbf{C}, W, \mathbf{X})\{Y - \mu^2(\mathbf{C}, W, \mathbf{X})\}$, $\phi^1((\mathbf{C}, W, X_1); \check{\mu}^2, \mu^1, \omega^1) := \omega^1(\mathbf{C}, X_1)\{\check{\mu}^2(\mathbf{C}, W, X_1) - \mu^1(\mathbf{C}, X_1)\}$, *and* $\phi^0(\mathbf{C}; \check{\mu}^1) := \check{\mu}^1(\mathbf{C}) - \psi_0$. *For* $i = 0, 1, 2$, *define* $\phi^i_0$ *as* $\phi^i$ *equipped with true nuisances* $(\mu^i_0, \pi^i_0)$ *and* $\hat{\phi}^i_\ell$ *as* $\phi^i$ *equipped with estimated nuisances* $\hat{\mu}^i_\ell, \hat{\pi}^i_\ell$. *Define* $R_i := (1/L) \sum_{\ell=1}^{L} (\mathbb{E}_{\mathcal{D}^i_\ell}[\hat{\phi}^i_\ell] - \mathbb{E}_{P^i}[\hat{\phi}^i_\ell])$ *for* $i = 0, 1, 2$. *Then,*

1. *The error* $\hat{\psi} - \psi_0$ *is decomposed as follows:*

$$\hat{\psi} - \psi_0 = \sum_{i=0}^{2} R_i + \frac{1}{L} \sum_{\ell=1}^{L} \sum_{i=1}^{2} \mathbb{E}_{P^i_{\pi_i}}[\{\hat{\mu}^i_\ell - \mu^i_0\}\{\omega^i_0 - \hat{\omega}^i_\ell\}]. \tag{4}$$

2. *Let* $\rho^2_{i,0} := \mathbb{V}_{P^i_{\pi_i}}[\phi^i_0]$. *With probability (W.P) greater than* $1 - \epsilon$,

$$\sum_{i=0}^{2} R_i \leqslant 3\sqrt{\frac{2}{\epsilon}} \left( \sqrt{\sum_{i=0}^{2} \frac{\rho_{i,0}}{|\mathcal{D}^i|}} + \sqrt{\sum_{\ell=1}^{L} \sum_{i=0}^{2} \frac{\|\hat{\phi}^i_\ell - \phi^i_0\|^2_{P^i_{\pi_i}}}{|\mathcal{D}^i_\ell|}} \right). \tag{5}$$

3. *Let* $\kappa^3_{i,0} := \mathbb{E}_{P^i_{\pi_i}}[|\phi^i_0|^3]$. *Let* $\Phi(x)$ *denote the standard normal CDF. W.P greater than* $1 - \epsilon$,

$$\left| P^i_{\pi_i} \left( \frac{\sqrt{|\mathcal{D}^i|}}{\rho_{k,0}} R_i < x \right) - \Phi(x) \right| \leqslant \frac{1}{\sqrt{2\pi}} \sqrt{\frac{L^2}{\epsilon} \sum_{\ell=1}^{L} \frac{\|\hat{\phi}^i_\ell - \phi^i_0\|^2_{P^i_{\pi_i}}}{|\mathcal{D}^i_\ell|}} + \frac{0.4748 \kappa^3_{i,0}}{\rho^3_{i,0} \sqrt{|\mathcal{D}^k|}}. \tag{6}$$

If the nuisance parameters $\hat{\mu}^i_\ell$ and $\hat{\pi}^i_\ell$ converge at a rate of $n^{-1/4}$ (where $n$ is the size of the smallest sample set), the DML estimator achieves a faster convergence rate of $n^{-1/2}$. This rapid convergence allows its asymptotic distribution to closely approximate the standard normal distribution, as is further clarified in the asymptotic analysis:

**Theorem 3** (**Asymptotic Error**). *Suppose each nuisance estimates* $\hat{\mu}^2_\ell, \hat{\mu}^1_\ell, \hat{\omega}^2_\ell, \hat{\omega}^1_\ell$ *are* $L_2$-*consistent and bounded. Then, the error of the DML estimator* $\hat{\psi}$ *in Def.* 5 *is given as follows:*

$$\hat{\psi} - \psi_0 = \sum_{i=0}^{2} R_i + \sum_{\ell=1}^{L} O_{P^2_{\pi_2}}(\|\hat{\mu}^2_\ell - \mu^2_0\|\|\hat{\omega}^2_\ell - \omega^2_0\|) + \sum_{\ell=1}^{L} O_{P^1_{\pi_1}}(\|\hat{\mu}^1_\ell - \mu^1_0\|\|\hat{\omega}^1_\ell - \omega^1_0\|), \tag{7}$$

*where* $R_i$ *converges in distribution to* $\mathtt{normal}(0, \rho^2_{i,0})$.

Eq. (7) implies that the error term $\hat{\psi} - \psi_0$ converges to zero faster than the convergence rate of nuisances, which is a property known as *debiasedness*.

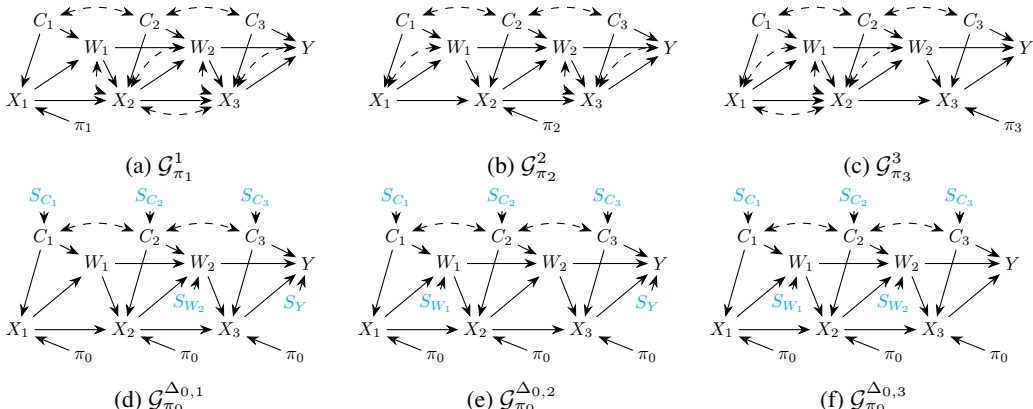

Figure 2: Graphs illustrating the inference of a multiple treatment strategy $\pi_0 := \{\pi_0(x_1 \mid c_1), \pi_0(x_2 \mid x_1, c_2, w_1), \pi_0(x_3 \mid x_2, c_3, w_2)\}$ given data from source domains $\mathcal{M}^1, \mathcal{M}^2, \mathcal{M}^3$.

# 4 Combining multiple experiments

In this section, we extend our method to incorporate the combination of data from multiple experiments, specifically focusing on $m$ different experiments derived from varied policies ($\pi_i$) in distinct source domains ($\mathcal{M}^i$). A practical scenario for this task is the following:

**Example 3** (Multi-stage treatment strategies). Consider a scenario involving hospitals in three different cities: New York (domain 1 with $P^1, \mathcal{G}^1$), Los Angeles (domain 2 with $P^2, \mathcal{G}^2$), and San Francisco (domain 3 with $P^3, \mathcal{G}^3$). Each hospital has different guidelines, i.e., policies, for diabetes treatment. In New York, the hospital focuses on insulin therapy adjustment based on the patient lifestyle choices, primarily for Type 1 Diabetes patients ($\pi_1$). In Los Angeles, the hospital focuses on team diet and exercise regimen adjustments, primarily for Type 2 Diabetes patients ($\pi_2$). In contrast, San Francisco's approach involves advanced monitoring and AI-driven predictive adjustments for higher-risk diabetes patients. Now, as the leader of a new clinical team in Chicago (the target domain with $P^0, \mathcal{G}^0$), the task is to evaluate a novel candidate treatment policy $\pi_0$, which integrates the strategies from these three domains to provide comprehensive care for both Type 1 and Type 2 Diabetes patients. The structure of the problem is captured in causal diagrams in Fig. 2, illustrating the data-generating process, the experiments in each city, and the assumed discrepancies between these source domains and Chicago. ∎

## 4.1 Identification

We consider a sequence of variables $(\mathbf{C}, X_1, W_1, \cdots, X_m, W_m := Y)$ where $(\mathbf{C}, \mathbf{W}^{(i-1)})$ represent the covariates corresponding to each of the $i$'th experiments, and $(X_1, \cdots, X_m)$ are the corresponding treatment variables. We are given samples drawn from $P^i_{\pi_i}(\mathbf{V})$ for $i = 1, \cdots, m$ and $P^0(\mathbf{C})$. We will leverage causal diagrams $\mathcal{G}_{\pi_i}$ and selection diagrams $\mathcal{G}^{\Delta_{0,i}}$ for every $i = 1, \cdots, m$. Formally, the task signature is given as follows:

- **Input**: Samples from $P^i_{\pi_i}(\mathbf{V})$ for $i = 1, \cdots, m$ and $P^0(\mathbf{C}_1, \mathbf{C}_2)$; Causal diagrams $\mathcal{G}^i_{\pi_i}$ and selection diagrams $\mathcal{G}^{\Delta_{0,i}}_{\pi_0}$ for $i = 1, \cdots, m$.

- **Query**: Estimate the effect of the target policy $\pi_0$ on the target domain $\mathcal{M}^0$; i.e., $\mathbb{E}_{P^0_{\pi_0}}[Y]$.

**Definition 6** (**Adjustment criterion for combining multiple experiments**). *The adjustment criterion for combining multiple policies are the following d-separation criterion in the the DTRs $\mathcal{G}_{\pi_0}, \mathcal{G}_{\pi_1}, \cdots, \mathcal{G}_{\pi_m}$ and the selection diagram $\mathcal{G}^{\Delta_{0,1}}_{\pi_0}, \cdots, \mathcal{G}^{\Delta_{0,m}}_{\pi_0}$.*

1. ***Domain transfer for*** $Y$: $(Y \perp\!\!\!\perp \mathbf{S} \mid \mathbf{C}, \mathbf{W}, \mathbf{X})$ *in* $\mathcal{G}^{\Delta_{0,m}}_{\pi_0}$*; i.e., the distribution over $Y$ is invariant between the source distribution from $\mathcal{M}^m$ and the target.*

2. ***Domain transfer for*** $W_i$ *for* $i = 1, \cdots, m-1$: $(W_i \perp\!\!\!\perp \mathbf{S} \mid \mathbf{C}^{(i)}, \mathbf{W}^{(i-1)})$ *in* $\mathcal{G}^{\Delta_{0,i}}_{\pi_0}$*; i.e., the distribution over $W_i$ is invariant between the source distribution from $\mathcal{M}^i$ and the target.*

3. **Adjustment for $Y$:** $(Y \perp\!\!\!\perp \pi_i \mid \mathbf{C}, \mathbf{W}, \mathbf{X})$ *in* $\mathcal{G}_{\pi_i}$ *for* $i \in \{0, m\}$; *i.e., the distribution over $Y$ is invariant between regimes $\pi_0$ and $\pi_m$.*

4. **Adjustment for $W_i$** $i = 1, \cdots, m-1$: $(W_i \perp\!\!\!\perp \pi_j \mid \mathbf{C}^{(i)}, \mathbf{X}^{(i-1)})$ *in* $\mathcal{G}_{\pi_j}$ *for* $j \in \{0, i\}$; *i.e., the distribution over $W_i$ is invariant between regimes $\pi_0$ and $\pi_i$.*

These conditions lead to the following identification criterion.

**Theorem 4** (**Adjustment for combining multiple experiments**). *Under the adjustment criterion in Def. 6, the target query $\psi_0 := \mathbb{E}_{P_{\pi_0}^0}[Y]$ is identifiable from the samples from $P_{\pi_1}^1(\mathbf{V}), \cdots, P_{\pi_m}^m(\mathbf{V})$ and $P^0(\mathbf{C})$. Specifically, it's expressed as follows:*

$$\mathbb{E}_{P_{\pi_0}^0}[Y] = \sum_{\mathbf{w}, \mathbf{c}, \mathbf{x}} \mathbb{E}_{P_{\pi_m}^m}[Y \mid \mathbf{c}, \mathbf{w}, \mathbf{x}] \prod_{i=1}^{m-1} P_{\pi_i}^i(w_i \mid \mathbf{c}, \mathbf{x}^{(i-1)}, \mathbf{w}^{(i-1)}) \prod_{j=1}^{m-1} \pi_0(x_j \mid \mathbf{c}, \mathbf{w}^{(j-1)}) P^0(\mathbf{c}),$$
$$(8)$$

### 4.2 Estimation

The regression nuisance parameters are defined as follows. We first define the following nuisance.

$$\mu_0^m(\mathbf{C}, \mathbf{W}, \mathbf{X}) := \mathbb{E}_{P_{\pi_m}^m}[Y \mid \mathbf{C}, \mathbf{W}, \mathbf{X}] \tag{9}$$

$$\check{\mu}_0^m(\mathbf{C}, \mathbf{W}, \mathbf{X}^{(m-1)}) := \sum_{x_m} \pi_0^m(x_m \mid \mathbf{C}, \mathbf{W}^{(m-1)}) \mu_0^m(\mathbf{C}, \mathbf{W}, \mathbf{X}^{(m-1)}, x_m) \tag{10}$$

For $i = m-1, \cdots, 1$, the other nuisances are defined in a following manner:

$$\mu_0^i(\mathbf{C}, \mathbf{W}^{(i-1)}, \mathbf{X}^{(i)}) := \mathbb{E}_{P_{\pi_i}^i}[\check{\mu}_0^{i+1}(\mathbf{C}, \mathbf{W}^{(i)}, \mathbf{X}^{(i)}) \mid \mathbf{C}, \mathbf{W}^{(i-1)}, \mathbf{X}^{(i)}], \tag{11}$$

$$\check{\mu}_0^i(\mathbf{C}, \mathbf{W}^{(i-1)}, \mathbf{X}^{(i-1)}) := \sum_{x_i} \mu_0^i(\mathbf{C}, \mathbf{W}^{(i-1)}, \mathbf{X}^{(i-1)}, x_i) \pi_0^i(x_i, \mathbf{C}, \mathbf{W}^{(i-1)}). \tag{12}$$

We note that Eq. (8) can be parameterized as $\mathbb{E}_{P_{\pi_0}^0}[Y] = \mathbb{E}_{P^0}[\check{\mu}_0^1(\mathbf{C})]$. On the other hand, the ratio nuisance parameters $\omega_0^i$ for $i = 1, \cdots, m$ are defined as functionals satisfying the followings:

$$\mathbb{E}_{P_{\boldsymbol{\pi}}^0}[Y] = \mathbb{E}_{P_{\pi_i}^i}[\mu_0^i(\mathbf{C}, \mathbf{W}^{(i-1)}, \mathbf{X}^{(i)}) \omega_0^i(\mathbf{C}, \mathbf{W}^{(i-1)}, \mathbf{X}^{(i)})], \tag{13}$$

where the closed form is given as

$$\omega_0^i = \frac{\pi_0(X_i \mid \mathbf{C}, \mathbf{W}^{(i-1)}) \prod_{j=1}^{i-1} P_{\pi_j}^j(W_j \mid \mathbf{C}, \mathbf{X}^{(j-1)}, \mathbf{W}^{(j-1)}) \pi_0(X_j \mid \mathbf{C}, \mathbf{W}^{(j-1)}) P^0(\mathbf{C})}{P_{\pi_0}^0(\mathbf{C}, \mathbf{W}^{(i-1)}, \mathbf{X}^{(i)})}$$
$$(14)$$

Eq. (8) can be parameterized as $\mathbb{E}_{P_{\pi_m}^m}[\omega^{(m)}(\mathbf{C}, \mathbf{W}^{(m-1)}, \mathbf{X}^{(m)}) Y]$. Equipped with these nuisances, we define a corresponding estimator as follows.

**Definition 7** (**DML for combining multiple experiments**). *Let $\mathcal{D}^i \sim P_{\pi_i}^i(\mathbf{V})$ for $i = 1, \cdots, m$ and $\mathcal{D}^0 \sim P^0(\mathbf{C})$. Let $L \geq 2$ denote a fixed number.*

1. **Sample split:** *For $\ell = 1, \cdots, L$, randomly split $\mathcal{D}^i$ for $i \in \{0, 1, \cdots, m\}$ into L-fold. The $\ell$'th partition of the sample is denoted $\mathcal{D}_\ell^i$. The complement is $\mathcal{D}_{-\ell}^i := \mathcal{D}^i \backslash \mathcal{D}_\ell^i$.*

2. **Nuisance estimation:** *For each $\ell = 1, \cdots, L$, learn the estimator model $\hat{\mu}_\ell^m, \cdots, \hat{\mu}_\ell^1$ for $\mu_0^m, \cdots, \mu_0^1$ using samples $\mathcal{D}_{-\ell}^m, \mathcal{D}_{-\ell}^1$, respectively. Also, learn the estimation model for $\hat{\omega}_\ell^1, \cdots, \hat{\omega}_\ell^m$ for $\omega_0^1, \cdots, \omega_0^m$ using samples $\mathcal{D}_{-\ell}^i$ for $i = 0, 1, \cdots, m$, respectively.*

3. **Evaluation:** *The DML estimator $\hat{\psi}$ for $\mathbb{E}_{P_{\pi_0}^0}[Y]$ is then given as*

$$\hat{\psi} := \frac{1}{L} \sum_{\ell=1}^{L} \sum_{i=1}^{m} \mathbb{E}_{\mathcal{D}_\ell^i}[\hat{\omega}_\ell^i \{\hat{\check{\mu}}_\ell^{i+1} - \hat{\mu}_\ell^i\}] + \mathbb{E}_{\mathcal{D}_\ell^0}[\check{\mu}_\ell^1]. \tag{15}$$

The error of the DML estimator is presented below:

**Theorem 5** (**Learning Guarantees**). *Suppose $\mu_0^i, \hat{\mu}_\ell^i < \infty$ and $0 < \pi_0^i, \hat{\pi}_\ell^i < \infty$ almost surely for $i = 1, \cdots, m$. Define $\phi^i((\mathbf{C}, \mathbf{W}^{(i)}, \mathbf{X}^{(i)}); \omega^i, \breve{\mu}^{i+1}, \mu^i) := \omega^i\{\breve{\mu}^{i+1} - \mu^i\}$ for $i = 1, \cdots, m$, where $\breve{\mu}^{m+1} := Y$. Let $\phi^0(\mathbf{C}; \breve{\mu}^1) := \breve{\mu}^1 - \psi_0$. For $i = 0, \cdots, m$, define $\phi_0^i$ as $\phi^i$ equipped with true nuisances, and $\hat{\phi}_\ell^i$ as $\phi^i$ equipped with estimated nuisances. Define $R_i := (1/L)\sum_{\ell=1}^L (\mathbb{E}_{\mathcal{D}_\ell^i}[\hat{\phi}_\ell^i] - \mathbb{E}_{P^i}[\hat{\phi}_\ell^i])$ for $i = 0, 1, \cdots, m$. Then,*

1. *The error $\hat{\psi} - \psi_0$ is decomposed as follows:*

$$\hat{\psi} - \psi_0 = \sum_{i=0}^m R_i + \frac{1}{L}\sum_{\ell=1}^L \sum_{i=1}^m \mathbb{E}_{P_{\pi_i}^i}[\{\hat{\mu}_\ell^i - \mu_0^i\}\{\omega_0^i - \hat{\omega}_\ell^i\}]. \tag{16}$$

2. *Let $\rho_{i,0}^2 := \mathbb{V}_{P_{\pi_i}^i}[\phi_0^i]$. With probability (W.P) greater than $1 - \epsilon$,*

$$\sum_{i=0}^m R_i \leqslant (m+1)\sqrt{\frac{2}{\epsilon}}\left(\sqrt{\sum_{i=0}^m \frac{\rho_{i,0}^2}{|\mathcal{D}^i|}} + \sqrt{\sum_{\ell=1}^L \sum_{i=0}^m \frac{\|\hat{\phi}_\ell^i - \phi_0^i\|_{P_{\pi_i}^i}^2}{|\mathcal{D}_\ell^i|}}\right). \tag{17}$$

3. *Let $\kappa_{i,0}^3 := \mathbb{E}_{P_{\pi_i}^i}[|\phi_0^i|^3]$. Let $\Phi(x)$ denote the standard normal CDF. W.P greater than $1 - \epsilon$,*

$$\left|P_{\pi_i}^i\left(\frac{\sqrt{|\mathcal{D}^i|}}{\rho_{k,0}}R_i < x\right) - \Phi(x)\right| \leqslant \frac{1}{\sqrt{2\pi}}\sqrt{\frac{1}{\epsilon}\sum_{\ell=1}^L \frac{\|\hat{\phi}_\ell^i - \phi_0^i\|_{P_{\pi_i}^i}^2}{|\mathcal{D}_\ell^i|}} + \frac{0.4748\kappa_{i,0}^3}{\rho_{i,0}^3\sqrt{|\mathcal{D}^k|}}. \tag{18}$$

A corresponding asymptotic error analysis is following:

**Theorem 6** (**Asymptotic Error**). *Suppose each nuisance estimates $\hat{\mu}_\ell^1, \cdots, \hat{\mu}_\ell^m$ and $\hat{\omega}_\ell^1, \cdots, \hat{\omega}_\ell^m$ are $L_2$-consistent and bounded. Then, the error of the DML estimator $\hat{\psi}$ in Def. 7 is given as follows:*

$$\hat{\psi} - \psi_0 = \sum_{i=0}^m R_i + \sum_{\ell=1}^L \sum_{i=1}^m O_{P_{\pi_i}^i}(\|\hat{\mu}_\ell^i - \mu_0^i\|\|\hat{\omega}_\ell^i - \omega_0^i\|), \tag{19}$$

*where $R_i$ converges in distribution to $\texttt{Normal}(0, \rho_{i,0}^2)$.*

Similarly to Thm. 3, this result implies that the DML estimator $\hat{\psi}$ converges fast even when the nuisance estimates converge relatively slowly.

## 5 Experiments

In this section, we demonstrate the proposed estimators in Defs. (5,7) for combining multiple experimental datasets from different domains. We first compared the estimators on synthetic data to provide evidence of the fast convergence and doubly robustness behaviours of the proposed estimators. We conclude with an analysis of the ACTG 175 clinical trial [21] and Project STAR. We will use $T^{\text{est}}(\mathbf{x})$ for est $\in \{\text{reg, pw, dml}\}$ to denote the estimators $\{\text{OM, PW, DML}\}$ for the policy effect $\mathbb{E}_{P_{\pi_0}^0} Y$. OM and PW estimators are purely based on the regression-based nuisances $\boldsymbol{\mu}$ and $\boldsymbol{\omega}$, respectively. To assess the quality of each estimator, we consider the *absolute error* (AE) as $\text{AE}^{\text{est}} = |T^{\text{est}}(\mathbf{x}) - \mathbb{E}_{P_{\pi_0}^0}[Y]|$. We used XGBoost [12] to estimate nuisances.

**Synthetic Simulations** We ran 100 simulations for each $N = \{2500, 5000, 10000, 20000\}$ where $N$ is the sample size. We measure the $\text{AE}^{\text{est}}$ in the presence of the 'converging noise $\epsilon$' in estimating the nuisance, decaying at a $N^{-1/4}$ rate (i.e., $\epsilon \sim \text{normal}(N^{-1/4}, N^{-1/4})$, where $N$ is the size of samples). To enforce the convergence rate of nuisance estimates no faster than the decaying rate $n^{-1/4}$, we add $\epsilon$ to all nuisance estimates. This scenario is inspired by the experimental design discussed in [27]. The AE plots for combining two/multiple experiments are presented in Figs. (3a, 3b). For all examples, the proposed DML estimator outperforms the other two estimators by achieving fast convergence. This result corroborates the robustness property in Thm. (3, 6), which implies that the proposed estimator converges faster than the other counterparts.

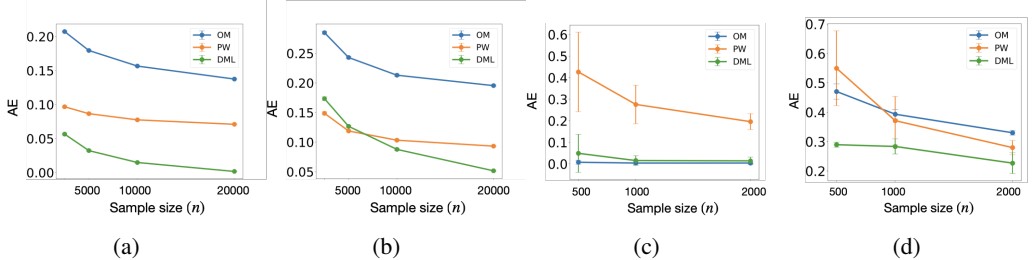

|     |     |     |     |
| :-: | :-: | :-: | :-: |
| (a) | (b) | (c) | (d) |

Figure 3: Comparison of the proposed DML estimator with other counterparts (outcome-based model called 'OM', and the probability-weighting-based model labelled 'PW') for **(a,b)** synthetic data analysis for combining two and multiple experiments; and **(c,d)** real-world data analysis under the noise-free or noisy environments in learning nuisances.

**External validity: ACTG 175**   To provide empirical evidence, we analyze the ACTG 175 random-ized trial [21], which assessed therapies for reducing CD4 T cell counts in HIV patients. Participants were randomly assigned to treatments $X_2 \in \{0, 1\}$, with prior anti-retroviral drug use $X_1 \in \{0, 1\}$ recorded. Patient demographics $C_1, C_2$—including gender, age, weight, and Karnofsky score—were collected, and CD4 T cell counts ($W$) were measured. To simulate an alternative study with a modi-fied guideline for $X_1$, we sub-sampled ACTG 175, adjusting covariate distributions and assignments of $\{X_1, X_2\}$. Specifically, we evaluate a stochastic policy $\pi_0 = \{\pi_0(x_1 \mid c_1), \pi_0(x_1 \mid c_2)\}$ for combining $X_1$ and $X_2$ based on $\mathbf{C}_1, \mathbf{C}_2$, with distribution $P^0$ representing a location with differing covariate distributions and treatment assignments. Further details are provided in Appendix D.2.

We evaluated the $\text{AE}^{\text{est}}$ of all proposed estimators with and without noise (as described in the synthetic simulations). The AE plots are shown in Figs. (3c, 3d). Results indicate that both the regression and DML estimators converge to the true policy effect faster under noisy conditions, whereas the PW estimator converges more slowly. However, DML does not consistently outperform at all sample sizes (see Fig. 3c), as its error is influenced by the combined errors in the OM and PW estimators. Consequently, high error in the PW estimator may lead to increased error in the DML estimator.

**External validity: Project STAR**   We further examine policies on teacher-student ratios (i.e., class sizes) to improve academic achieve-ment, using a semi-synthetic adaptation of the Project STAR dataset [40]. This longitudinal study evaluated the impact of teacher-student ratios on academic outcomes for students in kindergarten through third grade, with students randomized each year to one of three class size interventions. Here, we assess a 3-stage policy setting student-teacher ratios across Grades 0, 1, and 2, observing academic scores as interme-diate outcomes, with baseline covariates (e.g., ethnicity, gender) and final academic scores at the end of Grade 3 as the primary outcome. To

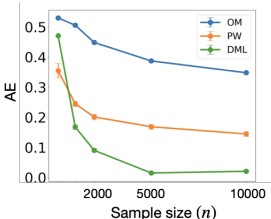

Figure 4: STAR Results.

emulate data collected across different domains, we subsample using various probabilities to shift baseline covariate distributions, as done in ACTG 175 (see Appendix D.3 for details). We evaluated the PW, OM, and DML estimators across dataset sizes, plotting their absolute errors against the true effect of the candidate policy. Results, shown in Fig. 4, mirror earlier experiments, with all estimators improving as sample size increases and DML showing faster convergence.

## 6   Conclusion

This paper has considered the evaluation of the effectiveness of policies in settings where the available data is sampled from distributions that differ from the population in the target domain. We have illustrated this task with the problem of extrapolating the results of a clinical trial in both working examples and real-world scenarios to evaluate variations of the treatment in different populations. Our contributions are (1) introducing several identification criteria for the effectiveness of policies given experimental datasets from two or more domains and (2) developing doubly robust estimators for these settings that achieve fast convergence.

## Acknowledgments

We thank anonymous reviewers for constructive comments to improve the manuscript.

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

# Supplement to "Efficient Policy Evaluation Across Multiple Different Experimental Datasets"

## Contents

# A  Related Work

Evaluating the impact of a policy using observational and experimental data under different conditions is a widespread challenge in various important decision-making fields. Formulations of this problem have appeared in the causality literature, but also in statistics, reinforcement learning, and epidemiology.

Off-policy evaluation (OPE) aims to assess the performance of policies of interest using observational samples. In this line of research, [44] considers generalizing the effect of a policy under distribution shift. We build on this intuition, but instead seek to *combine* multiple (policy-)interventional data from source domains to learn the effect of policies of interest on the target domain. [1, 23] for instance used auxiliary datasets from multiple bandit instance, though they setting assume that the datasets are sampled from the same underlying populations and environments. Several authors have also considered transfer learning in off-policy learning in the context of bandits [46, 5]. Further, [22] addresses the problem of selection biases in observational data for off-policy learning.

In the causal inference literature, combining multiple experimental studies to estimate a new causal effect is a task called generalized identification [30]. Recent progress has been made in developing corresponding estimators [26, 24]. However, these estimators are not applicable when our goal is to combine multiple policy interventional studies from source domains to estimate a causal effect in the target domain. Accordingly, [4, 29, 15] developed the notion of generalized transportability that aims to evaluate a causal effect on a target domain from multiple observational and / or interventional distributions from other source domains. In this line of research, our work relates closely to Correa and Bareinboim's identification algorithm for the effect of policies [15]. We similarly develop identification criteria that are conducive to efficient estimation from finite samples. In particular, our work focuses on the derivation of sample-efficient estimators for the policy effect of interest on the target domain.

From this perspective, our work can be interpreted as a bridge between causal inference and off-policy evaluation [34] since we leverage formal theories in causal inference (e.g., generalized identification [29], generalized transportability [15, 30]) to solve off-policy evaluation problems efficiently from finite samples. There are prior works that similarly integrated both fields. For instance, standard policy evaluation methods in the RL literature use the backdoor adjustment to learn the Q value as a function of the state, to address confounding effects [41]. Meanwhile, other studies have applied the front-door adjustment formula for OPE in the presence of unmeasured confounders [39]. Finally, some works have leveraged double negative controls for OPE [43].

# B  Broader Impact Statement

Our work investigates the conditions under which policies may be estimated from multipe datasets collected under different conditions. In this work, we start from the assumption that causal and selection diagrams that are consistent with the underlying data generating systems of interest are available. In general, this requires domain knowledge and should be justified by prior knowledge or experiment. It is important also to make the distinction between the task of partial identification, that is inferring an expression for bounds on causal effects, and that of estimation, that is providing efficient estimators from finite samples to compute bounds in practice. This set of results concerns mostly the second task. In higher-dimensional systems, the computational complexity of estimating the conditional expectations and density ratios that define our estimators could be a substantial challenge. Consequently, practitioners must exercise caution when deploying the proposed method in small sample scenarios where estimators may be inaccurate. Moreover, we have stated our convergence guarantees in the infinite sample limit, without quantifying the finite-sample estimation uncertainty. Finally, we emphasize that simulations on real and synthetic data are provided for illustration purposes only. These results do not recommend or advocate for the implementation of a particular policy, and should be considered in practice in combination with other aspects of the decision-making process.

## C  Proofs

### C.1  Proof for Theorem 1 and Theorem 4

Since Theorem 1 is a special case for Theorem 4, we will only prove for Theorem 4.

Note

$$\mathbb{E}_{P_{\pi_0}^0}[Y] = \sum_{\mathbf{w},\mathbf{c},\mathbf{x}} \mathbb{E}_{P_{\pi_0}^0}[Y \mid \mathbf{c},\mathbf{w},\mathbf{x}] \prod_{i=1}^{m-1} P_{\pi_0}^0(w_i \mid \mathbf{c}, \mathbf{X}^{(i-1)}, \mathbf{w}^{(i-1)}) \prod_{j=1}^{m-1} \pi_0(x_j \mid \mathbf{c}, \mathbf{w}^{(j-1)}) P^0(\mathbf{c}). \tag{1}$$

Then,

$$\mathbb{E}_{P_{\pi_0}^0}[Y \mid \mathbf{c},\mathbf{w},\mathbf{x}] = \mathbb{E}_{P_{\pi_2}^0}[Y \mid \mathbf{c},\mathbf{w},\mathbf{x}] \tag{2}$$
$$= \mathbb{E}_{P_{\pi_2}^2}[Y \mid \mathbf{c},\mathbf{w},\mathbf{x}], \tag{3}$$

by leveraging the domain transfer for $Y$ and adjustment for $Y$ condition.

For each $P_{\pi_0}^0(w_i \mid \mathbf{c}, \mathbf{X}^{(i-1)}, \mathbf{w}^{(i-1)})$,

$$P_{\pi_0}^0(w_i \mid \mathbf{c}, \mathbf{X}^{(i-1)}, \mathbf{w}^{(i-1)}) \tag{4}$$
$$= P_{\pi_0}^i(w_i \mid \mathbf{c}, \mathbf{X}^{(i-1)}, \mathbf{w}^{(i-1)}) \tag{5}$$
$$= P_{\pi_i}^i(w_i \mid \mathbf{c}, \mathbf{X}^{(i-1)}, \mathbf{w}^{(i-1)}), \tag{6}$$

again, by leveraging the domain transfer condition for $W_i$ and adjustment condition for $W_i$. This completes the proof.

### C.2  Proof for Theorem 2 and Theorem 5

Since Theorem 2 is a special case for Theorem 5, we will only prove for Theorem 5. Throughout the proof, we will use $\mathbf{C}_1 := \mathbf{C}$, $\mathbf{X}_i := \{X_i\}$ for $i = 1, \cdots, m$, and $\mathbf{C}_i := \{W_{i-1}\}$ for $i = 2, \cdots, m-1$. Also, we will sometimes use $P^1(\mathbf{C}_1) := P^0(\mathbf{C})$, $P^i(\mathbf{C}_i \mid \mathbf{C}^{(i-1)} \cup \mathbf{X}^{(i-1)})$ for $i > 1$ as $P_{\pi^{i-1}}^{i-1}(W_{i-1} \mid \mathbf{C}, \mathbf{W}^{(i-2)}, \mathbf{X}^{(i-1)})$.

#### C.2.1  Proof of Mixed Bias Property

Using the fact that $\psi_0 = \mathbb{E}_{P^0}[\check{\mu}_0^1]$, we can write it as

$$\psi_0 := \sum_{i=1}^m \underbrace{\mathbb{E}_{P_{\pi_i}^i}[\phi_0^i]}_{=0} + \mathbb{E}_{P^0}[\phi_0^0] = \sum_{i=0}^m \mathbb{E}_{P_{\pi_i}^i}[\phi_0^i]. \tag{7}$$

Then, we will claim and prove the following:

**Lemma 1** (**Mixed Bias Property**). *Suppose* $\mu_0^i, \hat{\mu}^i < \infty$ *and* $0 < \pi_0^i, \hat{\pi}^i < \infty$ *almost surely for* $i = 1, \cdots, m$. *For* $i = 1, 2, \cdots, m$, *define*

$$\phi^i((\mathbf{C}, \mathbf{W}^{(i)}, \mathbf{X}^{(i)}); \omega^i, \check{\mu}^{i+1}, \mu^i) := \omega^i \{\check{\mu}^{i+1} - \mu^i\}, \tag{8}$$

*and* $\check{\mu}^{m+1} := Y$. *Define* $\phi^0(\mathbf{C}; \check{\mu}^1) := \check{\mu}^1$. *For* $i = 0, \cdots, m$, *define* $\phi_0^i$ *as* $\phi^i$ *equipped with true nuisances, and* $\hat{\phi}^i$ *as* $\phi^i$ *equipped with estimated nuisances. Then,*

$$\sum_{i=0}^m \mathbb{E}_{P_{\pi_i}^i}[\hat{\phi}^i - \phi_0^i] = \sum_{i=1}^m \mathbb{E}_{P_{\pi_i}^i}[\{\hat{\mu}^i - \mu_0^i\}\{\omega_0^i - \hat{\omega}^i\}]. \tag{9}$$

*Proof of Lemma 1.* For $i = m, \cdots, 1$ with $\check{\mu}^{m+1} := Y$, define

$$\mu_0^i[\check{\mu}^{i+1}] := \mathbb{E}_{P_{\pi_i}^i}[\check{\mu}^{i+1} \mid \mathbf{C}, \mathbf{W}^{(i)}, \mathbf{X}^{(i)}].$$

Then,

$$\mathbb{E}_{P_{\pi_m}^m}\left[\hat{\omega}^m\{\check{\mu}^{m+1} - \hat{\mu}^m\}\right] + \mathbb{E}_{P_{\pi_m}^m}\left[\omega_0^m\hat{\mu}^m\right] - \underbrace{\mathbb{E}_{P_{\pi_m}^m}\left[\omega_0^m\mu_0^m[\check{\mu}^{i+1}]\right]}_{=\psi_0}$$

$$= \mathbb{E}_{P_{\pi_m}^m}\left[\{\hat{\omega}^m - \omega_0^m\}\{\mu_0^m[\check{\mu}^{i+1}] - \hat{\mu}^m\}\right].$$

For $i = m-1, \cdots, 1$,

$$\mathbb{E}_{P_{\pi_i}^i}\left[\hat{\omega}^i\{\check{\mu}^{i+1} - \hat{\mu}^i\}\right] + \mathbb{E}_{P_{\pi_i}^i}\left[\omega_0^i\hat{\mu}^i\right] - \mathbb{E}_{P_{\pi_i}^i}\left[\omega_0^i\mu_0^i[\check{\mu}^{i+1}]\right]$$

$$= \mathbb{E}_{P_{\pi_i}^i}\left[\{\hat{\omega}^i - \omega_0^i\}\{\mu_0^i[\check{\mu}^{i+1}] - \hat{\mu}^i\}\right].$$

Also, the following holds:

$$\mathbb{E}_{P_{\pi_{i+1}}^{i+1}}\left[\omega_0^{i+1}\hat{\mu}^{i+1}\right] = \mathbb{E}_{P_{\pi_i}^i}\left[\omega_0^i\mu_0^i[\check{\mu}^{i+1}]\right].$$

Finally, $\mathbb{E}_{P_{\pi_1}^1}\left[\omega_0^1\hat{\mu}^1\right] = \mathbb{E}_{P_{\pi_1}^1}\left[\check{\mu}^1\right]$.

Therefore,

$$\sum_{i=1}^{m}\mathbb{E}_{P_{\pi_i}^i}\left[\hat{\omega}^i\{\check{\mu}^{i+1} - \hat{\mu}^i\}\right] + \mathbb{E}_{P_{\pi_i}^i}\left[\omega_0^i\hat{\mu}^i\right] - \mathbb{E}_{P_{\pi_i}^i}\left[\omega_0^i\mu_0^i[\check{\mu}^{i+1}]\right]$$

$$= \sum_{i=0}^{m}\mathbb{E}_{P_{\pi_i}^i}\left[\hat{\phi}^i - \phi_0^i\right]$$

$$= \sum_{i=1}^{m}\mathbb{E}_{P_{\pi_i}^i}\left[\{\hat{\omega}^i - \omega_0^i\}\{\mu_0^i[\check{\mu}^{i+1}] - \hat{\mu}^i\}\right].$$

$\square$

### C.2.2 Proof for Statement 1

Recall

$$\hat{\psi} := \frac{1}{L}\sum_{\ell=1}^{L}\sum_{i=1}^{m}\mathbb{E}_{\mathcal{D}_\ell^i}\left[\hat{\omega}_\ell^i\{\hat{\check{\mu}}^{i+1} - \hat{\mu}_\ell^i\}\right] + \mathbb{E}_{\mathcal{D}_\ell^0}\left[\check{\mu}_\ell^1\right] = \frac{1}{L}\sum_{\ell=1}^{L}\sum_{i=0}^{m}\mathbb{E}_{\mathcal{D}_\ell^i}\left[\hat{\phi}_\ell^i\right]. \tag{10}$$

From Eq. (7),

$$\psi_0 := \frac{1}{L}\sum_{\ell=1}^{L}\sum_{i=0}^{m}\mathbb{E}_{P_{\pi_i}^i}\left[\phi_0^i\right]. \tag{11}$$

Then, the error $\hat{\psi} - \psi_0$ can be decomposed into

$$\hat{\psi} - \psi_0 = \sum_{i=0}^{m}\mathbb{E}_{\mathcal{D}^i - P_{\pi_i}^i}\left[\phi_0^i\right] + \frac{1}{L}\sum_{\ell=1}^{L}\sum_{i=0}^{m}\mathbb{E}_{\mathcal{D}^i - P_{\pi_i}^i}\left[\hat{\phi}_\ell^i - \phi_0^i\right] + \frac{1}{L}\sum_{\ell=1}^{L}\sum_{i=0}^{m}\mathbb{E}_{P_{\pi_i}^i}\left[\hat{\phi}_\ell^i - \phi_0^i\right]. \tag{12}$$

Define

$$R_i := \mathbb{E}_{\mathcal{D}^i - P_{\pi_i}^i}\left[\phi_0^i\right] + \frac{1}{L}\sum_{\ell=1}^{L}\mathbb{E}_{\mathcal{D}^i - P_{\pi_i}^i}\left[\hat{\phi}_\ell^i - \phi_0^i\right].$$

Then, the error can be represented as

$$\hat{\psi} - \psi_0 = \sum_{i=0}^{m}R_i + \frac{1}{L}\sum_{\ell=1}^{L}\sum_{i=0}^{m}\mathbb{E}_{P_{\pi_i}^i}\left[\hat{\phi}_\ell^i - \phi_0^i\right].$$

By Lemma 1,

$$\hat{\psi} - \psi_0 = \sum_{i=0}^{m} R_i + \frac{1}{L} \sum_{\ell=1}^{L} \sum_{i=0}^{m} \mathbb{E}_{P_{\pi_i}^i}[\{\hat{\omega}_\ell^i - \omega_0^i\}\{\mu_0^i - \hat{\mu}_\ell^i\}].$$

### C.2.3 Proof for Statement 2

We will use the following results:

**Lemma 2** (**Combining concentration inequalities**). *Suppose $P(A_k > t) \leqslant b_k/t^2$ for $k = 1, \cdots, K$. Then,*

$$P\left(\sum_{k=1}^{K} A_k \leqslant tK\right) \geqslant 1 - \frac{1}{t^2} \sum_{k=1}^{K} b_k.$$

***Proof of Lemma 2.*** The event $\sum_{k=1}^{K} A_k \leqslant tK$ includes the case where $A_k < t$ for $k = 1, \cdots, K$. Therefore,

$$
\begin{aligned}
P\left(\sum_{k=1}^{K} A_k \leqslant tK\right) &\geqslant P\left(A_1 \leqslant t \text{ and } \cdots \text{ and } A_K \leqslant t\right) \\
&= 1 - P\left(A_1 > t \text{ or } \cdots \text{ or } A_K > t\right) \\
&\geqslant 1 - \sum_{k=1}^{K} P\left(A_k > t\right) \\
&\geqslant 1 - \sum_{k=1}^{K} \frac{b_k}{t^2}.
\end{aligned}
$$

$\square$

**Lemma 3** (**Stochastic Equicontinuity**). *Let $\mathcal{D} \overset{iid}{\sim} P$. Let $\mathcal{D} = \mathcal{D}_0 \cup \mathcal{D}_1$, where $n := |\mathcal{D}_0|$. Let $\hat{f}$ be a function estimated from $\mathcal{D}_1$. Then, in probability greater than $1 - \epsilon$ for any $\epsilon \in (0, 1)$,*

$$\mathbb{E}_{\mathcal{D}_0 - P}\left[\left|\hat{f} - f\right|\right] \overset{w.p\ 1-\epsilon}{<} \frac{\|\hat{f} - f\|_P}{\sqrt{n\epsilon}}, \tag{13}$$

*which implies that*

$$\mathbb{E}_{\mathcal{D}_0 - P}[|\hat{f} - f|] = O_P\left(\frac{\|\hat{f} - f\|_P}{\sqrt{n}}\right).$$

***Proof of Lemma 3.*** This proof is from [28, Lemma 2]. Since $\hat{f}$ is a function of $\mathcal{D}_1$, we will denote $\hat{f}_{\mathcal{D}_1}$. Define a following random variable of interest:

$$X := \mathbb{E}_{\mathcal{D}_0 - P}[\hat{f}_{\mathcal{D}_1} - f].$$

Then, the conditional expectation of $X$ given $\mathcal{D}_1$ is zero, since

$$\mathbb{E}_P\left[\frac{1}{n} \sum_{i=1}^{n} \hat{f}_{\mathcal{D}_1}(\mathbf{V}_i) \,\middle|\, \mathcal{D}_1\right] = \frac{1}{n} \sum_{i=1}^{n} \mathbb{E}_P[\hat{f}_{\mathcal{D}_1}(\mathbf{V}_i) \mid \mathcal{D}_1] = \frac{1}{n} \sum_{i=1}^{n} \mathbb{E}_P[\hat{f}_{\mathcal{D}_1}(\mathbf{V}) \mid \mathcal{D}_1] = \mathbb{E}_P[\hat{f}_{\mathcal{D}_1}(\mathbf{V}) \mid \mathcal{D}_1],$$

where the third equality holds by the independence of $\mathcal{D}_0$ and $\mathcal{D}_1$. Therefore,

$$
\begin{aligned}
\mathbb{E}_P[X \mid \mathcal{D}_1] &= \mathbb{E}_P[\mathbb{E}_{\mathcal{D}_0 - P}[\hat{f}_{\mathcal{D}_1} - f] \mid \mathcal{D}_1] \\
&= \mathbb{E}_P[\mathbb{E}_{\mathcal{D}_0}[\hat{f}_{\mathcal{D}_1} - f] \mid \mathcal{D}_1] - \mathbb{E}_P[\mathbb{E}_P[\hat{f}_{\mathcal{D}_1} - f] \mid \mathcal{D}_1] \\
&= \mathbb{E}_P[\mathbb{E}_P[\hat{f}_{\mathcal{D}_1} - f] \mid \mathcal{D}_1] - \mathbb{E}_P[\mathbb{E}_P[\hat{f}_{\mathcal{D}_1} - f] \mid \mathcal{D}_1] = 0.
\end{aligned}
$$

Also,

$$\mathbb{V}_P[X \mid \mathcal{D}_1] = \mathbb{V}_P[\mathbb{E}_{\mathcal{D}_0 - P}[\hat{f}_{\mathcal{D}_1} - f] \mid \mathcal{D}_1]$$
$$= \mathbb{V}_P[\mathbb{E}_{\mathcal{D}_0}[\hat{f}_{\mathcal{D}_1} - f] \mid \mathcal{D}_1]$$
$$= \frac{1}{n}\mathbb{V}_P[\hat{f}_{\mathcal{D}_1} - f \mid \mathcal{D}_1]$$
$$\leqslant \frac{1}{n}\|\hat{f}_{\mathcal{D}_1} - f\|_P^2.$$

By applying the (conditional-) Chevyshev's inequality,

$$P(|X - \mathbb{E}_P[X \mid \mathcal{D}_1]| \geqslant t \mid \mathcal{D}_1) \leqslant \frac{1}{t^2}\mathbb{V}_P[X \mid \mathcal{D}_1] \leqslant \frac{1}{nt^2}\|\hat{f}_{\mathcal{D}_1} - f\|_P^2.$$

Then,

$$P(|X| \geqslant t) = P(|X - \mathbb{E}_P[X \mid \mathcal{D}_1]| \geqslant t)$$
$$= \mathbb{E}_{P(\mathcal{D}_1)}[P(|X - \mathbb{E}_P[X \mid \mathcal{D}_1]| \geqslant t \mid \mathcal{D}_1)]$$
$$\leqslant \frac{1}{nt^2}\|\hat{f}_{\mathcal{D}_1} - f\|_P^2.$$

In other words, $X < t$ in probability greater than $1 - \frac{1}{nt^2}\|\hat{f}_{\mathcal{D}_1} - f\|_P^2$. If $t = \frac{\|\hat{f}_{\mathcal{D}_1} - f\|_P}{\sqrt{n\epsilon}}$, then $X < \frac{\|\hat{f}_{\mathcal{D}_1} - f\|_P}{\sqrt{n\epsilon}}$ in the probability greater than $1 - \epsilon$ for any $\epsilon \in (0, 1)$. $\qquad\square$

Here, we will study the finite sample behavior of

$$\sum_{i=0}^{m} R_i := \sum_{i=0}^{m} \mathbb{E}_{\mathcal{D}^i - P_{\pi_i}^i}[\phi_0^i] + \frac{1}{L}\sum_{\ell=1}^{L}\sum_{i=0}^{m} \mathbb{E}_{\mathcal{D}_\ell^i - P_{\pi_i}^i}[\hat{\phi}_\ell^i - \phi_0^i].$$

By Chevyshev's inequality,

$$Pr\left(\left|\mathbb{E}_{\mathcal{D}^i - P_{\pi_i}^i}[\phi_0^i]\right| > t\frac{\rho_{i,0}}{\sqrt{|\mathcal{D}^i|}}\right) < \frac{1}{t^2},$$

or equivalently,

$$Pr\left(\left|\mathbb{E}_{\mathcal{D}^i - P_{\pi_i}^i}[\phi_0^i]\right| > t\right) < \frac{1}{t^2}\frac{\rho_{i,0}^2}{|\mathcal{D}^i|}.$$

By Lemma 2,

$$Pr\left(\sum_{i=0}^{m}\left|\mathbb{E}_{\mathcal{D}^i - P_{\pi_i}^i}[\phi_0^i]\right| \leqslant (m+1)t_1\right) \geqslant 1 - \frac{1}{t_1^2}\sum_{i=0}^{m}\frac{\rho_{i,0}^2}{|\mathcal{D}^i|}.$$

By Lemma 3,

$$Pr\left(\left|\mathbb{E}_{\mathcal{D}_\ell^i - P_{\pi_i}^i}[\hat{\phi}_\ell^i - \phi_0^i]\right| > t_2\right) \leqslant \frac{1}{t_2^2}\frac{\|\hat{\phi}_\ell^i - \phi_0^i\|_{P_{\pi_i}^i}^2}{|\mathcal{D}_\ell^i|}.$$

By Lemma 2,

$$P\left(\frac{1}{L}\sum_{\ell=1}^{L}\sum_{i=0}^{m}\left|\mathbb{E}_{\mathcal{D}_\ell^i - P_{\pi_i}^i}[\hat{\phi}_\ell^i - \phi_0^i]\right| \leqslant (m+1)t_2\right) \geqslant 1 - \frac{1}{t_2^2}\sum_{\ell=1}^{L}\sum_{i=0}^{m}\frac{\|\hat{\phi}_\ell^i - \phi_0^i\|_{P_{\pi_i}^i}^2}{|\mathcal{D}_\ell^i|}.$$

Choose $t_1 := \sqrt{\frac{2}{\epsilon} \sum_{i=0}^{m} \frac{\rho_{i,0}^2}{|\mathcal{D}^i|}}$ and $t_2 := \sqrt{\frac{2}{\epsilon} \sum_{\ell=1}^{L} \sum_{i=0}^{m} \frac{\|\hat{\phi}_\ell^i - \phi_0^i\|_{P_{\pi_i}^i}^2}{|\mathcal{D}_\ell^i|}}$. Then, with a probability greater than $1 - \epsilon$,

$$
\sum_{i=0}^{m} R_i \leqslant (m+1) \left( \sqrt{\frac{2}{\epsilon} \sum_{i=0}^{m} \frac{\rho_{i,0}^2}{|\mathcal{D}^i|}} + \sqrt{\frac{2}{\epsilon} \sum_{\ell=1}^{L} \sum_{i=1}^{K} \frac{\|\hat{\phi}_\ell^k - \phi_0^k\|_{P^k}^2}{|\mathcal{D}_\ell^k|}} \right)
$$

$$
= (m+1) \sqrt{\frac{2}{\epsilon}} \left( \sqrt{\sum_{i=0}^{m} \frac{\rho_{i,0}^2}{|\mathcal{D}^i|}} + \sqrt{\sum_{\ell=1}^{L} \sum_{i=0}^{m} \frac{\|\hat{\phi}_\ell^i - \phi_0^i\|_{P_{\pi_i}^i}^2}{|\mathcal{D}_\ell^i|}} \right).
$$

### C.2.4 Proof for Statement 3

We will use the following result:

**Proposition 1** (**Berry–Esseen's inequality** [7, 19, 38]). *Suppose* $\mathcal{D} = \{X_1, \cdots, X_n\}$ *are independent and identically distributed random variables with* $\mathbb{E}_P[X_i] = 0$, $\mathbb{E}_P[X_i^2] = \sigma^2$ *and* $\mathbb{E}_P[|X_i|^3] = \kappa^3$. *Then, for all* $x$ *and* $n$,

$$
\left| P\left( \frac{\sqrt{n}}{\sigma_0} \mathbb{E}_\mathcal{D}[X] < x \right) - \Phi(x) \right| \leqslant \frac{0.4748 \kappa^3}{\sigma^3 \sqrt{n}}.
$$

By Lemma 3,

$$
Pr\left( \left| \mathbb{E}_{\mathcal{D}_\ell^i - P_{\pi_i}^i}[\hat{\phi}_\ell^i - \phi_0^i] \right| > t \right) \leqslant \frac{1}{t^2} \frac{\|\hat{\phi}_\ell^i - \phi_0^i\|_{P_{\pi_i}^i}^2}{|\mathcal{D}_\ell^i|}. \tag{14}
$$

By Lemma 2,

$$
Pr\left( \frac{1}{L} \sum_{\ell=1}^{L} \left| \mathbb{E}_{\mathcal{D}_\ell^i - P_{\pi_i}^i}[\hat{\phi}_\ell^i - \phi_0^i] \right| \leqslant t \right) \geqslant 1 - \frac{1}{t^2} \sum_{\ell=1}^{L} \frac{\|\hat{\phi}_\ell^i - \phi_0^i\|_{P_{\pi_i}^i}^2}{|\mathcal{D}_\ell^i|}. \tag{15}
$$

Define

$$
\Delta_i := \sqrt{\frac{1}{\epsilon} \sum_{\ell=1}^{L} \frac{\|\hat{\phi}_\ell^i - \phi_0^i\|_{P_{\pi_i}^i}^2}{|\mathcal{D}_\ell^i|}}.
$$

With a probability greater than $1 - \epsilon$,

$$
\frac{1}{L} \sum_{\ell=1}^{L} \left| \mathbb{E}_{\mathcal{D}_\ell^i - P_{\pi_i}^i}[\hat{\phi}_\ell^i - \phi_0^i] \right| \overset{\text{w.p } 1-\epsilon}{\leqslant} \Delta_i.
$$

Define

$$
A_i := \mathbb{E}_{\mathcal{D}^i - P_{\pi_i}^i}[\phi_0^i]
$$

$$
B_i := \frac{1}{L} \sum_{\ell=1}^{L} \mathbb{E}_{\mathcal{D}_\ell^i - P_{\pi_i}^i}[\hat{\phi}_\ell^i - \phi_0^i]
$$

$$
C_i := \frac{1}{L} \sum_{\ell=1}^{L} \left| \mathbb{E}_{\mathcal{D}_\ell^i - P_{\pi_i}^i}[\hat{\phi}_\ell^i - \phi_0^i] \right|.
$$

Then, $R_i = A_i + B_i$. Then,

$$Pr(R_i < x) \tag{16}$$
$$= Pr(A_i + B_i < x) \tag{17}$$
$$= Pr(A_i < x - B_i) \tag{18}$$
$$\leqslant Pr(A_i < x + C_i) \tag{19}$$
$$\overset{\text{w.p } 1-\epsilon}{\leqslant} Pr(A_i < x + \Delta_i). \tag{20}$$

Then,

$$|Pr(A_i < x + \Delta_i) - \Phi(x)| \tag{21}$$
$$= |Pr(A_i < x + \Delta_i) - \Phi(x + \Delta_i) + \Phi(x + \Delta_i) - \Phi(x)| \tag{22}$$
$$\leqslant |Pr(A_i < x + \Delta_i) - \Phi(x + \Delta_i)| + |\Phi(x + \Delta_i) - \Phi(x)| \tag{23}$$
$$\leqslant \frac{0.4748\kappa_0^3}{\rho_{i,0}^3 \sqrt{|\mathcal{D}^i|}} + |\Phi(x + \Delta_i) - \Phi(x)| \qquad \text{(Prop. 1)} \tag{24}$$
$$= \frac{0.4748\kappa_0^3}{\rho_{i,0}^3 \sqrt{|\mathcal{D}^i|}} + |\Phi'(x')\Delta_i| \qquad \text{(Mean-value theorem)} \tag{25}$$
$$\leqslant \frac{0.4748\kappa_0^3}{\rho_{i,0}^3 \sqrt{|\mathcal{D}^i|}} + \frac{1}{\sqrt{2\pi}}\Delta_i. \tag{26}$$

This completes the proof. ∎

### C.3 Proof for Theorem 3 and Theorem 6

By Cauchy-Schwartz' inequality,

$$\frac{1}{L}\sum_{\ell=1}^{L}\sum_{i=0}^{m}\mathbb{E}_{P_{\pi_i}^i}[\{\mu_0^i - \hat{\mu}_\ell^i\}\{\hat{\omega}_\ell^i - \omega_0^i\}] \leqslant \frac{1}{L}\sum_{\ell=1}^{L}\sum_{i=0}^{m}O_{P_{\pi_i}^i}\left(\|\mu_0^i - \hat{\mu}_\ell^i\|\|\omega_0^i - \hat{\omega}_\ell^i\|\right). \tag{27}$$

Given assumption, the upper bound in Eq. (18) converges at $1/\sqrt{|\mathcal{D}_\ell^i|}$ rate. Therefore, $R_i$ converges in distribution to `normal(0, `$\rho_{i,0}^2$`)`.

## D    Details of Simulations

### D.1    Data Generating Process for Synthetic Simulations

Codes corresponding to simulations are submitted as supplementary materials.

#### D.1.1    Synthetic Simulations for Fig. 3a

We define the following SCM. First, $U_{XW}, U_{X_1,X_2}, U_{X_2,W}, U_{X_2,Y}, U_{C_{1,1}}, U_{C_{1,2}}, U_{C_{2,1}}, U_{C_{2,2}}, U_W, U_Y \sim$ `normal(0, 1)`. Then,

$$C_1 := f_{C_1}(S) = 0.25SU_{C_{1,1}} + 0.1S + U_{C_{1,1}}$$
$$C_2 := f_{C_2}(S) = 0.25SU_{C_{2,1}} + 0.1S + U_{C_{2,2}}$$
$$X_1 := f_{X_1}(C_1, C_2, S) \sim \texttt{Bernouli}(\pi_{1,S}(C_1, C_2))$$
$$W := f_W(C_1, C_2, X_1, U_{X_1,W}, S) = \texttt{sigmoid}(0.25SU_W + 0.5U_{X_1,W} + 3X_1 + 0.5(C_1 + C_2))$$
$$X_2 := f_{X_2}(X_1, W, C_1, C_2, S) \sim \texttt{Bernouli}(\pi_{2,S}(C_1, C_2))$$
$$Y := f_Y(C_1, C_2, X_1, X_2, W, U_{X_2,Y}, S) = \texttt{sigmoid}(0.5(C_1 + C_2) + 2(X_1 + X_2) - 2 - 0.5W$$
$$+ 0.1U_{X_2,Y} + 0.25SU_W).$$

Also, for $S \neq 0$,

$$\pi_{1,0} = \texttt{sigmoid}(C_1 + C_2 - 2)$$
$$\pi_{1,S} = \texttt{sigmoid}(0.5(C_1 + C_2) - 1)$$
$$\pi_{2,0} = \texttt{sigmoid}(0.5(C_1 + C_2) + 2(2X_1 - 1) - 0.5W + 1)$$
$$\pi_{2,S} = \texttt{sigmoid}(C_1 + C_2 + 2X_1 - 1 + 0.5W - 1).$$

### D.1.2 Synthetic Simulations for Fig. 3b

We define the following SCM. First, $U_{X_1,X_2}, U_{X_2,X_3}, U_{W_1,X_1}, U_{W_1,X_2}, U_{W_2,X_2}, U_{W_2,X_3}, U_{C_1,C_2},$
$U_{C_2,C_3}, U_{X_3,Y}, U_{C_{1,1}}, U_{C_{1,2}}, U_{C_{2,1}}, U_{C_{2,2}}, U_{C_{3,1}}, U_{C_{3,2}}, U_{W_1}, U_{W_2}, U_Y \sim \texttt{normal}(0,1)$. Then,

$$C_1 := f_{C_1}(S) = 0.25 S U_{C_{1,1}} + 0.1S + U_{C_{1,2}} + U_{C_1,C_2}$$
$$C_2 := f_{C_2}(S) = 0.25 S U_{C_{2,1}} + 0.1S + U_{C_{2,2}} + U_{C_1,C_2} + U_{C_2,C_3}$$
$$C_3 := f_{C_3}(S) = 0.25 S U_{C_{3,1}} + 0.1S + U_{C_{3,2}} + U_{C_2,C_3}$$
$$X_1 := f_{X_1}(C_1, C_2, S) \sim \texttt{Bernouli}(\pi_{1,S}(C_1, C_2))$$
$$W_1 := f_{W_1}(\mathbf{C}, X_1, U_{W_1,X_1}, U_{W_1,X_2}, S) = \texttt{sigmoid}(0.25 S U_{W_1} + 0.5(C_1 + C_2 + C_3)$$
$$- 1 + 3X_1 + 0.5(U_{W_1,X_1} + U_{W_1,X_2}) + S)$$
$$X_2 := f_{X_2}(X_1, W_1, C_1, C_2, C_3, S) \sim \texttt{Bernouli}(\pi_{2,S}(X_1, W_1, C_1, C_2, C_3, S))$$
$$W_2 := f_{W_2}(\mathbf{C}, X_1, X_2, W_1, U_{W_2,X_2}, U_{W_2,X_3}, S) = \texttt{sigmoid}(0.25 S U_{W_2} + 0.5(C_1 + C_2 + C_3)$$
$$- 1 + 3(X_1 + X_2) + 0.5(U_{W_2,X_2} + U_{W_2,X_3}) + S)$$
$$X_3 := f_{X_3}(X_1, X_2, W_1, W_2, C_1, C_2, C_3, S) \sim \texttt{Bernouli}(\pi_{3,S}(X_1, X_2, W_1, W_2, C_1, C_2, C_3, S))$$
$$Y := f_Y(\mathbf{C}, \mathbf{X}, \mathbf{W}, U_{X_3,Y}, S) = \texttt{sigmoid}(0.5(C_1 + C_2 + C_3) + 2(X_1 + X_2 + X_3)$$
$$- 3 - 0.5(W_1 + W_2) + 0.1 U_{X_3,Y} + 0.25 S U_W + S).$$

Also, for $S \neq 0$,

$$\pi_{1,0} = \texttt{sigmoid}(C_1 + C_2 + C_3 - 2)$$
$$\pi_{1,S} = \texttt{sigmoid}(0.5(C_1 + C_2 + C_3) - 1)$$
$$\pi_{2,0} = \texttt{sigmoid}(0.5(C_1 + C_2 + C_3) + 2(2X_1 - 1) - 0.5W + 1)$$
$$\pi_{2,S} = \texttt{sigmoid}(C_1 + C_2 + C_3 + 2X_1 - 1 + 0.5W - 1)$$
$$\pi_{3,0} = \texttt{sigmoid}(0.25(C_1 + C_2 + C_3) + (2X_1 - 1) - 0.25W_1 + 1 + (2X_2 - 1) - 0.25W_2 + 1)$$
$$\pi_{3,S} = \texttt{sigmoid}(0.5(C_1 + C_2 + C_3) + 2(2X_1 - 1) + 0.25W_1 - 1 + 2(2X_2 - 1) + 0.25W_2 - 1).$$

### D.2 External validity of the ACTG 175 clinical trial

To provide empirical evidence of policy estimation in a real-world setting, we revisit the ACTG 175 randomized clinical trial from 1994 conducted on patients from the United States and Puerto Rico [21]. It investigated the effectiveness of different therapies for reducing CD4 T cell counts in individuals with HIV (selected subject to various inclusion criteria). In the study, individuals were randomly assigned to two different treatments $X_2 \in \{0, 1\}$, and a record was made on whether a previous anti-retroviral drug had been administered $X_1 \in \{0, 1\}$ prior to the start of the trial. Patient demographics $C_1, C_2$ including gender, age, weight, among others, were collected, and CD4 T cell count were measured at treatment time $W$, and again 20 weeks after treatment initialization $Y$, the outcome of the analysis. To simulate a second study with a different guideline for anti-retroviral drug administration, we considered a sub-sampled version of ACTG 175 in which covariate distributions as well as the assignment of $X_1, X_2$ were modified.

ACTG 175 is an experimental study in which $X_2$ has been randomized and $X_1$ follows a baseline, unknown, stochastic policy $\pi_2 : \Omega_{C_2} \times \Omega_{X_1} \to [0, 1]$ assumed to depend on study-specific features $C_2$ such a patient's Karnofsky score and symptomatic indicators (both normalized to lie in the $[0, 1]$ interval). Samples of variables $C_1, C_2, W, X_1, X_2, Y$ therefore follow a distribution $P^2_{\text{rand}(X_2),\pi_2}(C_1, C_2, W, X_1, X_2, Y)$. The suffix "rand$(X_2)$" denotes a policy that randomizes $X_2$, i.e. $X_2 \sim \text{Bern}(0.5)$.

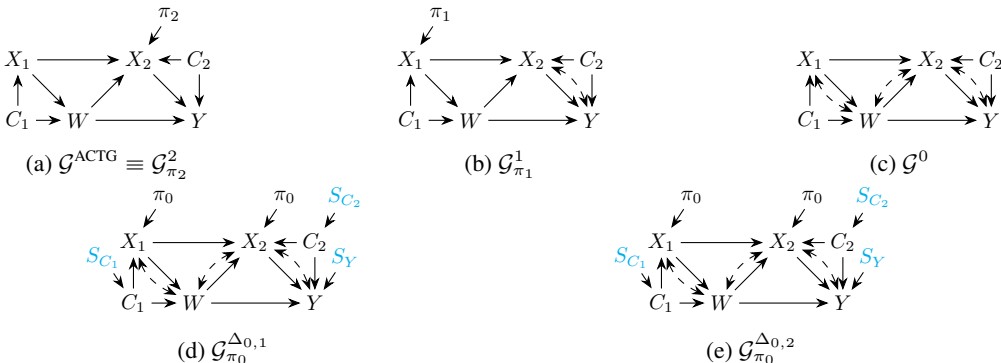

Figure 5: Causal diagrams and selection diagrams of the ACTG 175 experiment.

We generate a data sample from a second domain ($S = 1$) following the marginalized distribution $P^1_{\pi_1}(\boldsymbol{C}_1, W, X_1)$, mimicking a simple stochastic guideline on $X_1$ in which $\pi_1 := \pi_1(x_1 = 1 \mid \boldsymbol{c}_1) = 1/(1 + \exp\{-\boldsymbol{c}_{11} - \boldsymbol{c}_{12} - \boldsymbol{c}_{13}\})$. Higher values of $\boldsymbol{C}_1$ (taken to be normalized measurements of weight, height, and age) lead to higher likelihood of treatment. We achieve this dataset by sampling according to a re-weighted version of the ACTG 175 trial. In particular, we collect data from $P^1$ according to,

$$P^1_{\pi_1}(\boldsymbol{C}_1, W, X_1) := P^2_{\pi_2}(\boldsymbol{C}_1, W, X_1)\frac{\pi_1(X_1 \mid \boldsymbol{C}_1)P^1(\boldsymbol{C}_1)}{P^2_{\pi_2}(X_1 \mid \boldsymbol{C}_1)}$$

For this example, we consider evaluating a stochastic policy $\pi_0 = \{\pi_0(x_1 \mid \boldsymbol{c}_1), \pi_0(x_1 \mid \boldsymbol{c}_2)\}$ that combines the drugs $X_1, X_2$ according to a stochastic policy for $X_1$ based on weight, height, and age, ($\boldsymbol{C}_1$) and for $X_2$ based on a patient's Karnofsky score and symptomatic indicators ($\boldsymbol{C}_2$). In particular,

$$\pi_0(x_1 = 1 \mid \boldsymbol{c}_1) = 1/(1 + \exp\{-\boldsymbol{c}_{11} - 1\}), \quad \pi_0(x_2 = 1 \mid \boldsymbol{c}_2) = 1/(1 + \exp\{-0.5\boldsymbol{c}_{21} - \boldsymbol{c}_{22}\}). \tag{1}$$

The policy $\pi_0$ is considered to be implemented on a patient population located in a different location that are know to have a differing covariate distribution $P^0(\boldsymbol{C}_1, \boldsymbol{C}_2)$ to that observed in ACTG 175 and the second study, among other discrepancies. We assume that the SCM generating this experimental study follows the causal graphs in Fig. 5.

The target population under $\pi_0$ is then given by

$$P^0_{\pi_0}(\boldsymbol{C}_1, \boldsymbol{C}_2, W, X_1, X_2, Y)$$
$$:= P^2_{\text{rand}(X_2), \pi_2}(\boldsymbol{C}_1, \boldsymbol{C}_2, W, X_1, X_2, Y)\frac{P^0(\boldsymbol{C}_1, \boldsymbol{C}_2)\pi_0(X_1 \mid \boldsymbol{C}_1)\pi_0(X_2 \mid \boldsymbol{C}_2)}{P^2(\boldsymbol{C}_1, \boldsymbol{C}_2)P^2_{\pi_2}(X_1 \mid \boldsymbol{C}_1)P^2_{\text{rand}(X_2), \pi_2}(X_2)}$$

We limit all datasets to approximately 2000 samples as this is the size of the ACTG 175 trial. The ground truth target effect $\mathbb{E}_{P^0_{\pi_0}}[Y]$ is evaluated by taking the empirical mean of $Y$ in the sample of data collected from $P^0$ with the procedure above.

### D.3 External validity of the Project STAR study

We describe in this section additional experimental details on the Project STAR study[4] . This study investigated the impact of teacher/student ratios on academic achievement for kindergarten through third-grade students. Project STAR was a four-year longitudinal study where students were randomly assigned to one of three interventions with different class sizes each year, following different randomization procedures. The causal diagram we assume for this setting is provided in

---

[4]The dataset is publicly accessible from the R data repository: https://search.r-project.org/CRAN/refmans/AER/html/STAR.html.

Fig. 6. Bi-directed arcs denote unobserved confounding in the observational regime (when student are observed in a particular class size rather than forced to join a particular class size).

Specifically, we consider the evaluation of a 3-stage stochastic policy,

$$\pi_0 = \{\pi_0(x_1 \mid c_1), \pi_0(x_2 \mid c_2, x_1, w_1), \pi_0(x_3 \mid c_3, x_2, w_2)\},$$

where,

$$\pi_0(x_1 \mid c_1) := 1/(1 + \exp\{c_{11} + c_{12} - 2\})$$
$$\pi_0(x_2 \mid c_2, x_1, w_1) := 1/(1 + \exp\{0.5(c_{21} + c_{22}) + 2(2x_1 - 1) - 0.5w_1 + 1\})$$
$$\pi_0(x_3 \mid c_3, x_2, w_2) := 1/(1 + \exp\{0.5(c_{31} + c_{32}) + (2x_1 - 1) - 0.25w_1 + (2x_2 - 1) - 0.25w_2 + 1\})$$

These policies determine the student-teacher ratio $X_0, X_1, X_2$, taking values "regular" or "small", across three different grades, namely Grade 0 (Kindergarten), Grade 1 and Grade 2. $C$ refers to a two-dimensional demographic variable encoding gender and ethnicity, converted to binary and categorical variables respectively. (To avoid clutter, in Fig. 6 we use $C = C_1 = C_2 = C_3$.) $W_1, W_2$ are intermediate school outcomes that include the sum total of an individual's reading score and math score in grades 0 (Kindergarten) and 1 respectively. $Y$ is the outcome of interest and represents total reading score and math score in grade 2.

To mimic the setting where data at different stages was collected from different domains, we subsample the dataset using different sets of probabilities to induce differences in the distributions of baseline covariates. In particular, we fix the dataset in the target domain ($S = 0$) to the distribution observed in the study and sub-sample according to different probabilities to create datasets for domains $S = 1, S = 2$, and $S = 3$, as follows.

We generate a sample of data from a first source domain ($S = 1$) following the marginalized distribution $P_{\pi_1}^1(C_1, W_1, X_1)$, where $\pi_1 := \pi_1(x_1 = 1 \mid c_1) = 1/(1 + \exp\{0.5(c_{11} + c_{12}) - 1\})$ defines the probability for the student-teacher ratio variables in Kindergarten in domain $S = 1$. We achieve this dataset by sampling according to a re-weighted version of the STAR study. In particular, we collect data from $P^1$ according to,

$$P_{\pi_1}^1(C_1, W_1, X_1) := P^0(C_1, W, X_1)\frac{\pi_1(X_1 \mid C_1)P^1(C_1)}{P^0(X_1 \mid C_1)}$$

where $P^1(c_1) = 0.3$ if $c_{11} = 1, c_{12} = 1$ and $P^1(c_1) = 0.7$ otherwise.

We generate a sample of data from a second source domain ($S = 2$) following the marginalized distribution $P_{\pi_2}^2(C_2, W_1, W_2, X_1, X_2)$, where $\pi_2 := \pi_2(x_2 = 1 \mid c_2, x_1, w_1) = 1/(1 + \exp\{0.5(c_{21} + c_{22}) + 2x_1 - 1 - 0.5w_1 - 1\})$ defines the probability for the student-teacher ratio variables in grade 1 in domain $S = 2$. We achieve this dataset by sampling according to a re-weighted version of the STAR study. In particular, we collect data from $P^2$ according to,

$$P_{\pi_2}^2(C_2, W_1, W_2, X_1, X_2) := P^0(C_2, W_1, W_2, X_1, X_2)\frac{\pi_2(X_2 \mid C_2, X_1, W_1)P^2(C_2)}{P^0(X_2 \mid C_2, X_1, W_1)}$$

where $P^2(c_2) = 0.7$ if $c_{21} = 1, c_{22} = 1$ and $P^2(c_2) = 0.3$ otherwise.

We generate a sample of data from a third source domain ($S = 3$) following the marginalized distribution $P_{\pi_3}^3(C_3, W_1, W_2, X_1, X_2, X_3)$, where $\pi_3 := \pi_3(x_3 = 1 \mid c_3, x_1, w_1, x_2, w_2) = 1/(1 + \exp\{0.5(c_{31} + c_{32}) + 2(2x_1 - 1) - 0.25w_1 + (2x_2 - 1) + 0.25w_2 - 1\})$ defines the probability for the student-teacher ratio variables in grade 3 in domain $S = 3$. We achieve this dataset by sampling according to a re-weighted version of the STAR study. In particular, we collect data from $P^3$ according to,

$$P_{\pi_3}^3(C_3, W_1, W_2, X_1, X_2, X_3, Y) :=$$
$$P^0(C_3, W_1, W_2, X_1, X_2, X_3, Y)\frac{\pi_3(X_3 \mid C_3, W_1, W_2, X_1, X_2, X_3)P^3(C_3)}{P^0(X_3 \mid C_3, X_1, W_1, X_2, W_2)}$$

where $P^3(c_3) = 0.5$ if $c_{31} = 1, c_{32} = 1$ and $P^3(c_3) = 0.5$ otherwise.

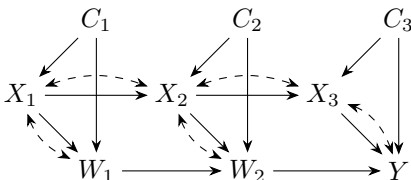

Figure 6: Causal diagram assumed for the target domain of the STAR Project study. To avoid cluttering the diagram we write $C = C_1 = C_2 = C_3$, i.e., all $C$'s refer to the same variables (gender and ethnicity).

